# Vulnerability to memory decline in aging revealed by a mega-analysis of structural brain change

Brain atrophy is a key factor behind episodic memory loss in aging, but the nature and ubiquity of this relationship remains poorly understood. This study leverages 13 longitudinal datasets, including 3737 cognitively healthy adults (10,343 MRI scans; 13,460 memory assessments), to determine whether brain change-memory change associations are more pronounced with age and genetic risk for Alzheimer's Disease. Both factors are associated with accelerated brain decline, yet it remains unclear whether memory loss is exacerbated beyond what atrophy alone would predict. Additionally, we assess whether memory decline aligns with a global pattern of atrophy or stems from distinct regional contributions. Our mega-analysis reveals a nonlinear relationship between memory decline and brain atrophy, primarily affecting individuals with above-average brain structural decline. The associations are stronger in the hippocampus but also spread across diverse cortical and subcortical regions. The associations strengthen with age, reaching moderate associations in participants in their eighties. While APOE ε4 carriers exhibit steeper brain and memory loss, genetic risk has no effect on the change-change associations. These findings support the presence of common biological macrostructural substrates underlying memory function in older age which are vulnerable to multiple age-related factors, even in the absence of overt pathological changes.

Episodic memory declines with age[1,2], although individual trajectories vary significantly, with some experiencing marked cognitive decline and others maintaining relatively stable memory function over time[3]. Brain atrophy is considered a key contributor to these changes[4,5]. However, key questions remain poorly understood such as whether the change – change associations are dependent on age and genetic risk for Alzheimer's disease (AD) or if the patterns are driven exclusively by a subset of individuals with severe atrophy. Further, extant research remains inconclusive on whether the effects reflect global patterns of brain atrophy[6,7] or regional structural vulnerabilities, particularly in the hippocampus[8,9]. To examine these questions, we leveraged 13 datasets with more than 3700 cognitively healthy adult participants who underwent repeated MRI scans and cognitive assessments, together with simulations to guide results interpretation.

Both episodic memory and medial temporal lobe (MTL) structures tend to show relative stability across early adulthood and middle age in longitudinal studies, with a more marked decline from about 60 years[1,2,10–12]. In contrast, trajectories of cortical thickness outside the MTL tend to follow largely monotonic declines across adulthood, likely with subtle acceleration in thinning late in life[10,13–15], with a relatively large degree of heterogeneity for the remaining subcortical structures[11,16,17]. Aging is also associated with increased interindividual variability in cognition and biological measures, including memory performance and brain structure[16,18] and, even more importantly, the variability of brain and memory change also increases with age[19–21].

✉e-mail: d.v.pineiro@psykologi.uio.no

Initial evidence for a relationship between memory function and brain structure in cognitively healthy aging came from cross-sectional research[22,23], which indicated that differences in brain structure, particularly in the MTL, explained a modest amount of age-related variability in episodic memory function. Yet, it is now widely recognized that only longitudinal designs can effectively link co-occurring within-person changes in brain and cognition throughout the lifespan[24]. Indeed, longitudinal research has corroborated the association between memory decline and loss of gray matter in medial temporal features such as hippocampal atrophy[25–29] and entorhinal thinning[8,30]. The findings align with the key role of these structures in episodic memory and their susceptibility to aging and AD[31–33]. Outside this region, associations have also been reported, albeit less consistently, in the frontal, parietal, and temporal lobes[34,35], as well as in global gray matter[6,27]. These results are consistent with the complex cortico-subcortical circuitry supporting episodic memory function[33,36–38]. Debate remains on whether these change-change associations are driven by a main factor of brain decline or by one or more of these memory-sensitive structures[6,26,29,34,39]. Current research shows evidence of both domain-general dimensions of cortical and cognitive change[6,40] and domain-specific associations between MTL and episodic memory change[39].

Since age affects both individual trajectories of brain structure and memory, and their variability across individuals, it is likely that age also moderates their relationship, such that the associations strengthen with increasing age. However, limited data exist on this topic, and when available, this data primarily concerns the hippocampus. Cross-sectional evidence suggests stronger associations between hippocampus volume and memory performance in late life[41,42]. Using longitudinal data, Gorbach and colleagues[25] found an association between hippocampal atrophy and memory decline that was significant in older (65 – 80 years) but not in middle-aged (55 – 60 years) individuals, and further suggested that steeper declines in memory and hippocampal volume facilitate the detection of these associations.

The Apolipoprotein (APOE) ε4 allele represents the strongest known genetic risk factor for late-onset AD. In individuals with AD, ε4 carriers exhibit steeper brain atrophy, especially in limbic regions, and greater memory loss compared to non-carriers[43,44]. This pattern has also been observed in cognitively healthy older individuals[45,46], although not universally[47]. Some theoretical models propose distinct AD subtypes based on the APOE ε4 allele[48]. In this account, the ε4 carrier subtype represents a more severe, limbic-dominant form of AD, characterized by steeper loss of memory and stronger links to amyloid pathophysiology. The non-carrier subtype is thought to represent a milder form, more heterogeneous, and more associated with environmental factors[49]. This model also predicts stronger associations between memory change and atrophy, particularly in the MTL, in ε4 carriers. Supporting this, some studies report a stronger brain-memory association amongst APOE ε4 carriers, in AD samples[50] and, crucially, in a longitudinal sample of cognitively healthy older adults[9].

Not all observed changes in brain structure in older individuals reflect long-term changes, i.e., brain aging, but also short-term variations due to known, e.g., physical or cognitive training programs[51,52], and unknown factors, i.e., noise[20]. Hence, only a fraction of the observed changes occurring in brain structure over time may relate to memory decline. Thus, it is possible that most variation in brain structure over time is not degenerative, and that only individuals with severe atrophy show memory loss, leading to non-linear change–change associations.

Given the inherent heterogeneity in brain and cognitive trajectories, study samples, and analytical approaches – along with partially conflicting findings and the need to sample diverse populations for broad conclusions -, large-scale mega-analyses across cohorts are essential to accurately investigate change-change relationships

between episodic memory and structural brain decline[5]. Here, we conduct such a study to address key questions in the field: 1) Do brain change-memory change associations become more pronounced with increasing age? 2) Are these relationships consistent with a global factor of brain structural decline, MTL vulnerability, or multiple regional contributions? 3) Are change-change associations more pronounced in individuals with above-average brain decline and/or carriers of the APOE ε4 allele? We used a normative modeling framework to harmonize 13 datasets with longitudinal MRI scans and cognitive assessments, resulting in a final sample of 3737 cognitively healthy adults (10,343 MRI scans; 13,460 memory assessments) (Table 1, Supplementary Fig. 1). We estimated individual change in memory (Δmemory) and 166 brain cortical and subcortical regions (Δbrain), with a particular focus on the hippocampus. We focused on thickness as a measure of cortical change given its susceptibility to age[53,54], used a mega-analytical approach to maximize statistical power[55], and general additive mixed models (GAMMs) to enable greater analytical flexibility. Throughout the text, we refer to individuals with above- or below-average brain decline – relative to their age and sex peers – as brain decliners and maintainers. These labels do not reflect strict methodological criteria but are used solely for communicative clarity, as our analyses employ a dimensional approach throughout. Finally, we provide complete statistics, point estimates, and visualizations in a Supporting App (https://vidalpineiro.shinyapps.io/brain_mem_change/) and simulate data to aid the interpretation of results.

## Results

### Linking brain change and memory decline: main effects

We used GAMMs to assess the relationship between brain change (Δbrain) (cortical thickness and subcortical volume) and memory change (Δmemory) in 3737 adults. Δbrain was modeled as a smooth term, and the dataset as a random intercept. Sex and age trends were regressed out during normative modeling-based preprocessing and thus not included in the higher-level model. Henceforth, the change in the brain and memory for a given individual is relative to their age and sex peers. Control analyses with age, sex, or intracranial volume (ICV) as covariates, or with random slopes per dataset, did not substantially affect the outcome of the main results (Supporting App). Data were weighted to account for differences in reliability of change, as longitudinal data with fewer observations and shorter follow-up time contain more uncertainty[20]. Regions were defined based on the *Destrieux*[56] cortical and *aseg*[57] subcortical atlas within FreeSurfer. Nineteen regions showed significant, False Discovery Rate (FDR) corrected ($p_{FDR} < 0.05$) (Fig. 1a), mostly subcortical structures and temporal regions. The relationship for all these regions was non-linear, generally showing an association between Δbrain and Δmemory only when Δbrain was steeper than average (henceforth in *brain decliners*). When brain decline was milder than average (henceforth in *brain maintainers*), the association between Δbrain and Δmemory disappeared. Left and right hippocampus ($\beta_{weighted[w]}[l]$ = .165, effective degrees of freedom (*edf*) = 3.2, $p_{FDR} < 0.001$, *partial-[p]* $R^2 = 1.9\%$); $\beta_w[r] = 0.168$, *edf* = 3.3, $p_{FDR} < 0.001$, $_pR^2 = 1.6\%$), left amygdala ($\beta_w = 0.155$, *edf* = 4.5, $p_{FDR} < 0.001$, $_pR^2 = 1.1\%$), left thalamus ($\beta_w = 0.135$, *edf* = 3.7, $p_{FDR} = 0.03$, $_pR^2 = 0.8\%$), right long insular gyrus ($\beta_w = 0.135$, *edf* = 2.8, $p_{FDR} = 0.02$, $_pR^2 = 0.8\%$) and left parahippocampal gyrus ($\beta_w = 0.131$, $p_{FDR} = 0.02$, *edf* = 4.4, $_pR^2 = 0.8\%$) were amongst the regions showing strongest change – change associations in above-average brain decliners. See Fig. 1b for visualization of selected regions. See Supplementary Table 1 for statistics in significant regions. See Supporting App for complete statistics and visualization in all regions. No strong evidence for left–right asymmetry in change–change associations was found (Supplementary Fig. 2; SI). Education level did not significantly contribute either as a resilience factor, i.e., influencing the effect of education on brain and memory decline, or as a cognitive reserve factor, i.e., moderating the association between brain and

**Table 1 | Main Sample sociodemographics**

| Dataset | Subjects | Apoe ε4 | Age | Obs. Memory | | Obs. Brain | | Time Brain | Time Memory |
|---|---|---|---|---|---|---|---|---|---|
| | N (m) | NC:C | M (SD) range | N | M (SD) range | N | M (SD) range | M (SD) range | M (SD) range |
| adni | 223 (92) | 158:65 | 72.4 (6.3) | 1118 | 5.0 (2.7) | 984 | 4.4 (2.2) | 3.9 (2.1) | 4.6 (2.8) |
| | | | 55.8–89.9 | | 2–13 | | 2–12 | 1.6–9.6 | 2.0–13.5 |
| aibl | 142 (70) | 94:78 | 71.5 (6.4) | 564 | 4.0 (1.1) | 493 | 3.5 (1.1) | 4.2 (1.6) | 4.9 (1.6) |
| | | | 60.0–87.0 | | 2–5 | | 2–5 | 2.0–8.0 | 2.0–7.0 |
| base-ii | 214 (127) | 133:39 | 64.8 (15.0) | 572 | 2.7 (0.5) | 428 | 2.0 (0) | 2.3 (0.4) | 4.5 (1.5) |
| | | | 24.5–83.1 | | 2–3 | | 2–2 | 1.5–3.1 | 1.6–6.4 |
| bbhi | 256 (141) | 30:3 | 54.0 (7.2) | 512 | 2.0 (0) | 512 | 2.0 (0) | 2.4 (0.2) | 2.4 (0.2) |
| | | | 41.2–66.1 | | 2–2 | | 2–2 | 1.6–3.0 | 1.6–3.0 |
| betula | 47 (27) | 26:15 | 43.8 (12.8) | 94 | 2.0 (0) | 94 | 2.0 (0) | 4.3 (0.4) | 4.3 (0.4) |
| | | | 25.0–75.0 | | 2–2 | | 2–2 | 4.0–5.0 | 4.0–5.0 |
| cognorm | 93 (42) | 54:38 | 73.2 (6.0) | 596 | 6.4 (1.0) | 366 | 3.9 (1.3) | 6.0 (2.6) | 5.8 (0.9) |
| | | | 64.7–90.0 | | 3–7 | | 2–6 | 1.8–9.5 | 2.1–6.9 |
| habs | 223 (94) | 157:63 | 73.3 (6.1) | 1180 | 5.3 (1.0) | 615 | 2.8 (0.6) | 4.4 (1.2) | 4.5 (1.1) |
| | | | 62.5–89.3 | | 3–6 | | 2–4 | 2.0–8.5 | 2.0–8.5 |
| lcbc | 304 (120) | 100:72 | 47.9 (19.1) | 781 | 2.6 (0.7) | 799 | 2.6 (0.8) | 6.2 (2.8) | 6.0 (2.7) |
| | | | 20.0–85.5 | | 2–4 | | 2–7 | 2.4–11.5 | 2.4–11.5 |
| oasis3 | 431 (171) | 287:137 | 67.2 (8.9) | 3502 | 8.1 (4.0) | 1458 | 3.4 (1.5) | 6.4 (3.4) | 9.9 (4.2) |
| | | | 43.5–95.6 | | 2–23 | | 2–8 | 1.5–15.8 | 2.0–24.0 |
| preventad | 184 (53) | 115:69 | 63.6 (5.0) | 874 | 4.8 (1.0) | 1061 | 5.8 (1.0) | 3.3 (0.8) | 3.2 (0.8) |
| | | | 55.1–83.3 | | 3–6 | | 4–7 | 1.9–4.7 | 1.6–4.5 |
| Ub | 77 (28) | 67:10 | 68.6 (5.0) | 214 | 2.8 (0.4) | 208 | 2.7 (0.5) | 3.5 (1.0) | 3.7 (1.0) |
| | | | 51.7–78.1 | | 2–3 | | 2–3 | 1.6–5.0 | 1.6–5.2 |
| Ukb | 1066 (525) | 775:272 | 62.3 (7.0) | 2132 | 2.0 (0) | 2132 | 2.0 (0) | 2.3 (0.1) | 2.3 (0.1) |
| | | | 47.0–79.5 | | 2–2 | | 2–2 | 2.0–2.7 | 2.0–2.7 |
| vetsa | 477 (477) | 354:122 | 58.0 (3.9) | 1321 | 2.8 (0.4) | 1193 | 2.5 (0.5) | 9.3 (2.7) | 10.0 (2.4) |
| | | | 51.1–70.1 | | 2–3 | | 2–3 | 4.5–13.4 | 4.5–14.4 |
| All | 3737 (1967) | 2350:953 | 62.5 (11.6) | 13460 | 3.6 (2.6) | 10343 | 2.8 (1.3) | 4.5 (3.0) | 5.1 (3.6) |
| | | | 20.1–95.6 | | 2–23 | | 2–12 | 1.5–15.8 | 1.6–24.0 |

Main sociodemographic and observational detail of the main sample used in all main analyses. *N* Total number of individuals or observations, *NC* Non-carriers, *C* Carriers. *M* mean, *SD* Standard Deviation, *Obs* Observations.

memory, consistent with recent longitudinal studies on brain and memory decline[58,59] (SI).

### Age as a moderator of brain change - memory change associations

Next, we assessed whether the association between brain change and memory change varied with increasing age using tensor smooths (i.e., interactions between marginal smooth terms) as implemented in GAMM. For 7 regions, age significantly moderated the change - change associations ($p_{FDR} < .05$) (Fig. 2a) namely left ($p_{FDR} = 0.02$, $edf = 3.3$, $_pR^2 = 1.0\%$) and right hippocampus ($p_{FDR} = 0.02$, $edf = 4.2$, $_pR^2 = 1.0\%$), right inferior lateral ventricle ($p_{FDR} < 0.001$, $edf = 7.9$, $_pR^2 = 1.5\%$), left lateral ventricle ($p_{FDR} = 0.05$, $edf = 1.9$, $_pR^2 = 0.8\%$), right caudate ($p_{FDR} = 0.03$, $edf = 2.9$, $_pR^2 = 0.6\%$), right putamen ($p_{FDR} = 0.03$, $edf = 2.2$, $_pR^2 = 0.7\%$), and left short insular gyrus ($p_{FDR} = 0.02$, $edf = 2.8$, $_pR^2 = 0.7\%$). Combining the variance explained by the brain change and the brain change × age interaction regressors for the left hippocampus explained up to 2.9% of the variance in memory change. In most of these regions, we found that change–change associations increased with higher age and progressively included brain maintainers. Change – change associations in some regions begin to be apparent between 50 and 60 years. These regions differ in the specific shape of the interaction. For example, change – change associations in brain decliners are first apparent at ≈50 years in the right hippocampus, ≈60 years for the left hippocampus and the right lateral inferior ventricle,

and ≈70 years for the short insular gyrus and the right caudate. Similarly, associations between Δbrain and Δmemory in brain maintainers are apparent from ≈70 years in the left hippocampus and the right lateral inferior ventricle, but not in other regions such as the short insular gyrus or the right hippocampus. We use the left hippocampus to illustrate these effects: the relationship between Δbrain and Δmemory in brain decliners (i.e., point estimates) is $\beta_w = -0.04$ at age 40 years, $\beta_w = .02$ at 50 years, $\beta_w = 0.13$ at 60 years, $\beta_w = 0.23$ at 70 years, and $\beta_w = 0.29$ at 80 years. In contrast, the relationship between Δbrain and – Δmemory in brain maintainers is non-existent until age 70 ($\beta_w = .13$), with a small increase at age 80 ($\beta_w = .19$). See Fig. 2b for visualization of selected regions. See Supplementary Table 2 for statistics in significant regions and complete statistical outcomes in the Supporting App.

### Dimensionality of brain change

Next, we explored the dimensionality of those regional brain changes associated with memory decline. Is memory loss associated with a single global effect of brain decline, or does it reflect region-specific contributions? We computed the correlation of brain change across brain regions (Fig. 3a, mean $r = 0.14$ [0.10]; range = −0.04 −0.58) and carried out a PCA and a consensus clustering analysis to investigate this question. On the one hand, the PCA revealed that the first principal component (PC1) accounted for a somewhat modest 20.7% of the total variance, with all its loadings pointing in the same direction, and a

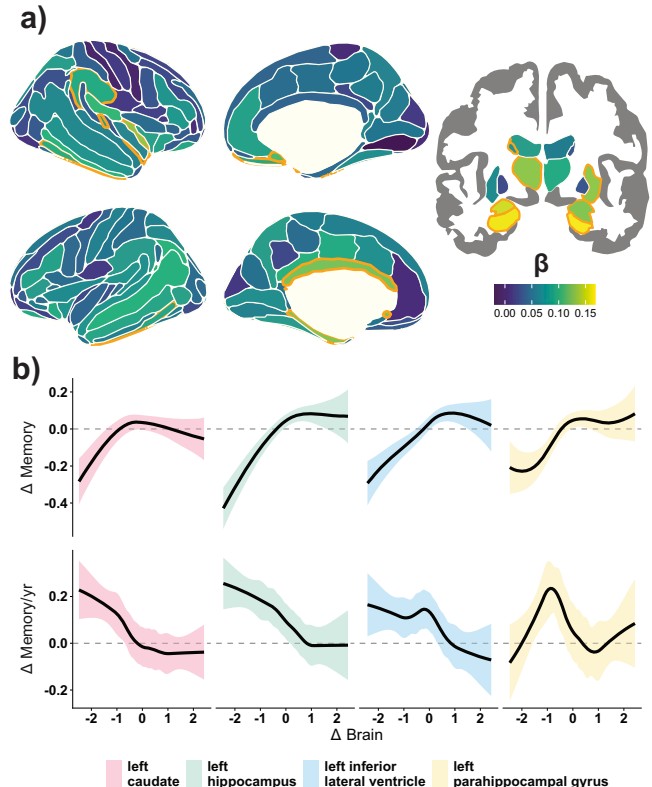

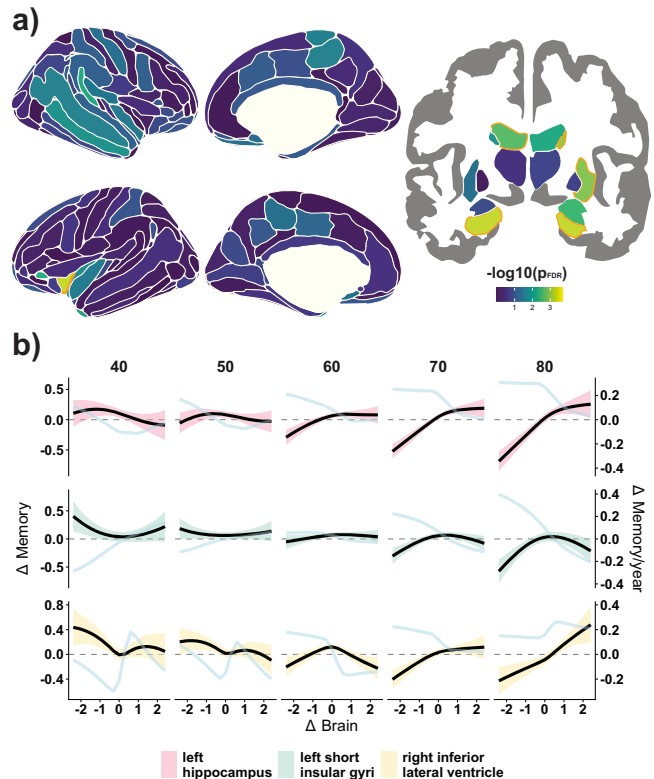

**Fig. 1 | Brain change – memory change associations. a** Estimates ($\beta_w$) for associations between Δbrain and Δmemory with orange color line representing $p_{FDR} < 0.05$. Δbrain represents atrophy in subcortical and thinning in cortical regions and is operationalized as a smooth variable in GAMMs. P-values were obtained via wild bootstrapping. **b** Change – change association for selected regions: left caudate ($\beta_w = .074$, $edf = 3.4$, $p_{FDR} = 0.029$, $_pR^2 = 0.6\%$, $n = 3730$), left hippocampus ($\beta_w = .165$, $edf = 3.2$, $p_{FDR} < 0.001$, $_pR^2 = 1.9\%$, $n = 3729$), left inferior lateral ventricle ($\beta_w = .132$, $edf = 3.2$, $p_{FDR} < 0.001$, $_pR^2 = 1.1\%$, $n = 3731$), and left parahippocampal gyrus ($\beta_w = .131$, $edf = 4.4$, $p_{FDR} = 0.02$, $_pR^2 = 0.8\%$, $n = 3731$). Upper plots display the smooth associations, while lower plots show their derivative (i.e., the association between Δbrain and Δmemory for each Δbrain value). Line and shaded ribbons represent the mean and 95% CIs. See Supplementary Table 1 and Supporting app for more information. $\beta_w$ = Density-weighted betas, using the derivative function. $edf$ = estimated degrees of freedom. $_pR^2$ = partial variance explained. $pFDR$ = False Discovered Rate-corrected p-values.

**Fig. 2 | Effect of age on Brain change – Memory change associations. a** $-\log10(p_{FDR})$ values of the effect of age on Δbrain – Δmemory associations modeled as GAMM-based tensor interaction terms. Orange color line represents $p_{FDR} < 0.05$. Δbrain represents atrophy in subcortical and thinning in cortical regions. P-values were obtained via wild bootstrapping. **b** Change – change association for selected regions at specific ages: left hippocampus ($edf = 3.3$, $p_{FDR} < 0.02$, $_pR^2 = 1.0\%$, $n = 3729$), left short insular gyri ($edf = 2.8$, $p_{FDR} = 0.02$, $_pR^2 = 0.7\%$, $n = 3729$), and right inferior lateral ventricle ($edf = 7.9$, $p_{FDR} < 0.001$, $_pR^2 = 1.5\%$, $n = 3725$). Black line - and colored 95% CIs ribbons - display mean Δbrain - Δmemory associations at a given age. The light blue line represents their derivative (i.e., association between Δbrain and Δmemory for each Δbrain value). See Supplementary Table 2 and Supporting app for more information. Note that Δbrain is derived from normative data and thus does not necessarily reflect the same amount of decline at each age. $edf$ estimated degrees of freedom. $_pR^2$ = partial variance explained. $pFDR$ False Discovered Rate-corrected p-values.

significant, ≈two-fold, fall in the variance explained by subsequent components (Fig. 3b). This suggests a pattern of brain change that to some degree aligns with the presence of a global pattern of brain decline.

On the other hand, consensus clustering analysis was performed to explore whether the effects were regional. Importantly, we tested using Monte-Carlo simulations whether the clustering solution rejected the null hypothesis of $K = 1$ cluster. Several clustering solutions rejected the null hypothesis, with 8 clusters being the best solution (Fig. 3c; Supplementary Fig. 3, Supplementary Table 3). Three clusters were subcortical (clusters #1 - #3) - one comprised of the left and right hippocampus and the left lateral ventricle - and 5 were cortical. Further, we assessed whether change in any of these clusters was associated with memory change, by respectively controlling for the hippocampus-based cluster, the main factor of brain decline, and by introducing all clusters together in a single model. Five clusters showed significant Δbrain – Δmemory associations controlling for the hippocampal-based cluster (cluster #1): cluster #3, the left and right amygdala ($\beta_w = .11$, $edf = 1.2$, $p = 0.01$); cluster #5, left pericallosal sulcus, right straight gyrus, and right subcallosal gyrus ($\beta_w = .16$, $edf = 3.1$, $p = 0.002$), cluster #6, right long insular gyrus and planum polare

($\beta_w = .06$, $edf = 1.0$, $p = 0.012$); cluster #7, right transverse temporal sulcus ($\beta_w = .05$, $edf = 3.0$, $p = 0.024$); and cluster #8, left parahippocampal gyrus ($\beta_w = .05$, $edf = 4.4$, $p = .0301$). Similar results were found when using the main component of brain decline and when all clusters were added in a single model. The regional model, including brain change in the eight regional clusters, explained 1.3% more variance in memory decline than the model including only the global pattern of brain decline ($partial$-$R^2 = 3.1\%$ and 1.9%, respectively). See SI and Supplementary Table 4 for detailed information. Altogether, the results suggest both global and regional factors influence the associations between brain and memory change.

## Influence of APOE ε4 status on brain change - memory change associations

A total of 3149 subjects had APOE data available. Of these, 27.8 % were carriers of the APOE ε4 allele (carriers vs. non-carriers). First, we assessed whether carriers of the APOE ε4 allele showed a steeper decline in brain or memory and whether this relationship was associated with age. For the main effects, we used linear mixed models with APOE ε4 allele as predictor and dataset as random intercept. For the interaction, we used GAMM with age as a smooth term by APOE ε4

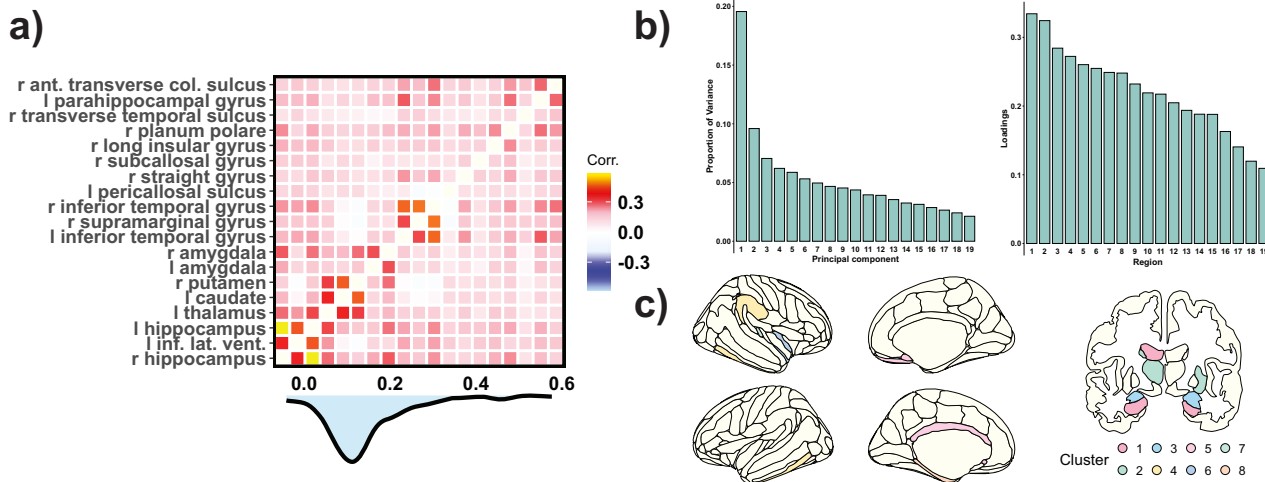

**Fig. 3 | Dimensionality of brain change. a** Pearson's r cross-correlation matrix of Δbrain across regions with significant Δbrain − Δmemory associations (N = 19, see Fig. 1). Below, a density plot of the correlation coefficients. **b** Variance explained by the principal PCA components and loading of the first component on the brain

regions showing significant Δbrain − Δmemory associations. See ID to region name correspondence in Supplementary Table 3. **c** Optimal consensus clustering solution (K = 8 clusters). Ant anterior; inf inferior; lat lateral; col collateral; l left; r right; vent ventricle.

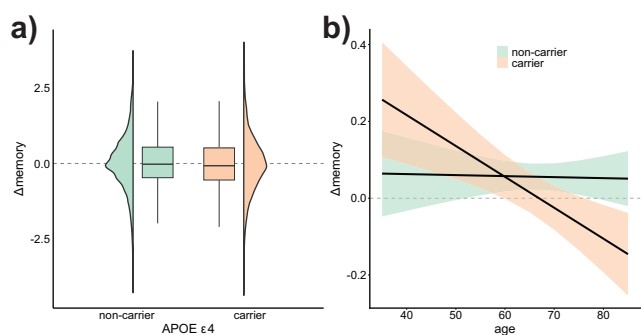

**Fig. 4 | APOE ε4 associations with memory change. a** Association between APOE ε4 (carriers vs. non-carriers) and memory change ($\beta[CI] = -0.035[-0.10, 0.02]$, $t[p] = -0.95[0.34]$, $n = 3298$) assessed using linear mixed models. Boxes represent the interquartile range (IQR), and whiskers extend to 1.5 × IQR. Line represents the median. Distribution of values outside whiskers can be assessed in the contiguous density plots. **b** Association between APOE ε4 (carriers, non-carriers) and memory change as a function of age ($\beta_w = -0.008$, $edf = 1$, $p = 0.03$, $n = 3298$). The interaction between APOE ε4 and age is operationalized as a smooth, factor interaction in a GAMM model. Line and ribbons represent mean and 95% CI. $\beta_w$ = Density-weighted betas, using the derivative function.

allele as an ordered factor. APOE ε4 was not significantly related to memory decline ($\beta[CI] = -0.035[-0.10, 0.02]$, $t[df] = -0.95[3296]$, $p = 0.34$) (Fig. 4a) but the relationship between APOE status and memory change increased with age ($\beta_w = -0.008$, $edf = 1$, $p = 0.03$), that is, APOE ε4 carriers showed less memory decline until ≈60 years of age, and more memory decline thereafter (Fig. 4b).

APOE ε4 was associated with steeper left and right hippocampal atrophy ($\beta_l[CI] = -0.135[-0.20, -0.07]$, $t[df] = -4.2[3228.1]$, $p_{FDR} = 0.004$; $\beta_r[CI] = -0.138[-0.20, -0.07]$, $t[df] = -4.1[3261.1]$, $p_{FDR} = 0.003$) as well as right amygdala atrophy ($\beta[CI] = -0.154[-0.22, -0.09]$, $t[df] = -4.7[3249]$, $p_{FDR} < 0.001$) (Fig. 5a). Considering all regions together, no global effect of the APOE ε4 allele on brain decline was found ($\beta[CI] = -0.004[-0.01, 0.00]$, $t[df] = -1.30[165]$, $p = 0.20$) (Fig. 5b). APOE ε4 was not significantly associated with steeper brain decline ($p_{FDR} > .10$) with higher age. However, at an uncorrected level, APOE ε4 was associated with a higher degree of hippocampal atrophy with increasing age ($\beta_{wl} = -0.011$, $edf = 1$,

$p_{unc} = 0.01$; $\beta_w r = -0.009$, $edf = 1$, $p_{unc} = 0.04$, respectively) (Supplementary Fig. 4).

Next, we assessed whether being an APOE ε4 carrier had any influence on the association between Δbrain and Δmemory. No region showed moderating effects of APOE ε4 on Δbrain − Δmemory associations ($p_{FDR} > .50$). APOE ε4 did not significantly moderate the relationship between Δbrain and Δmemory in the left and right hippocampus ($\beta_{wl} = -0.011$, $edf = 1$, $p_{unc} = 0.44$; $\beta_{wr} = -0.001$, $edf = 1$, $p_{unc} = 0.61$) (Fig. 6a). Finally, no regions showed a significant interaction between APOE ε4, age, and Δbrain on Δmemory ($p_{FDR} > .15$). Left and right hippocampus showed comparable Δbrain - Δmemory associations with age regardless of APOE ε4 status ($\beta_{wl} = .044$, $edf = 1$, $p_{unc} = 0.48$; $\beta_{wr} = -0.028$, $edf = 1$, $p_{unc} = 0.66$) (Fig. 6b).

We conducted two sensitivity analyses: I) In a subsample restricted to APOE ε4 non-carriers, where both the change−change effects and the moderator effects of age were comparable to those observed in the main sample (SI). 2) We reran the APOE ε4 analyses in a subsample of individuals aged 60 years or more at baseline ($n = 2048$). As expected, the effect of APOE ε4 status was greater on memory decline and found in a more brain regions. However, due to the restricted age range, APOE ε4 × age interactions were not significant. Consistent with the main analyses, APOE ε4 status did not moderate the relationship between brain change and memory change nor its interaction with age ($p_{FDR} > .15$) (SI; Supplementary Fig. 5). See complete statistics and additional visualization for all APOE ε4 analyses in the Supporting App. Altogether, the results show that cognitively healthy carriers of the APOE ε4 allele have steeper rates of brain and memory decline, specifically in old adulthood, but no evidence of stronger Δbrain − Δmemory associations. The regional associations between brain change and memory change exist independently of an increased presence of pathological processes and cognitive changes associated with the genetic risk of AD.

### Exploring mechanisms behind brain−cognitive relationships: A Post-hoc simulation study

Finally, we aimed to provide potential explanations for the empirical findings, focusing on the non-linear change−change associations, the moderating effect of age, and the absence of APOE ε4 effects. We used a simplified schematic model using the *sn* v.2.1.1 package[60], where observed brain data was the result of two underlying sources. The first source represented brain aging, characterized by a negatively skewed

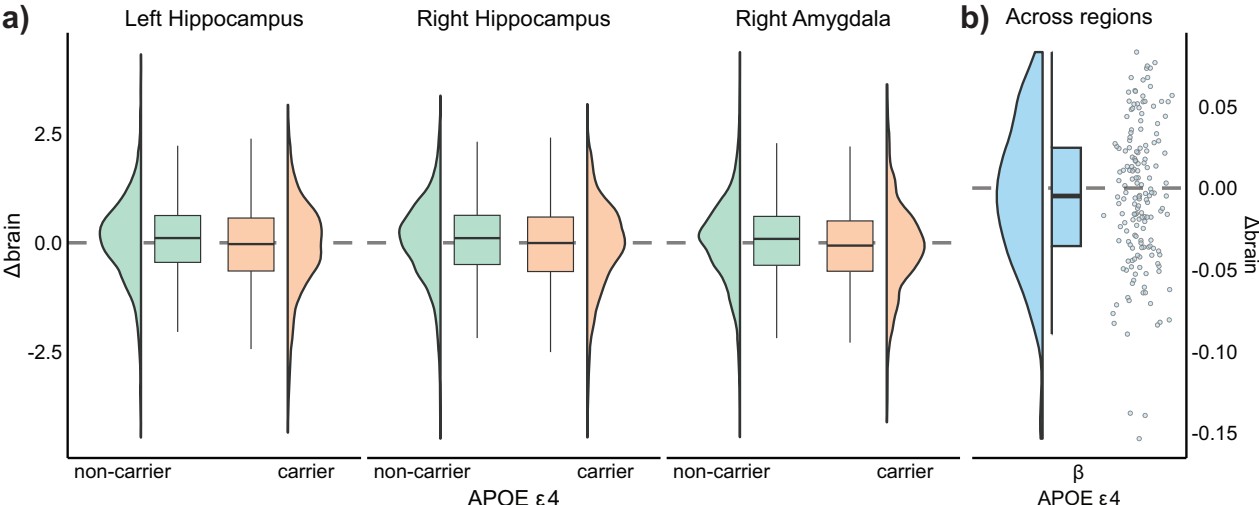

**Fig. 5 | APOE ε4 associations with brain change. a** Association between APOE ε4 (carriers vs. non-carriers) and brain change in the left and right hippocampus ($\beta_l[CI] = -0.135[-0.20,-0.07]$, $t(df) = -4.2(3228.1)$, $p_{FDR} = 0.004$, $n = 3310$; $\beta_r[CI] = -0.138[-0.20,-0.07]$, $t(df) = -4.1(3261.1)$, $p_{FDR} = 0.003$, $n = 3297$; and the right amygdala $\beta[CI] = -0.154[-0.22,-0.09]$, $t(df) = -4.7(3249)$, $p_{FDR} = .001$, $n = 3302$) assessed with linear mixed model. **b** Effect of APOE ε4 on brain change across all regions ($h_O = 0$; $\beta = -0.004$, $t(df) = -1.30(165)$, $p = .20$, $n = 166$) assessed using a one-sample t-test. Each point represents a region. Note that the three regions with more negative effects of APOE ε4 correspond to those displayed in panel a). Boxes represent the interquartile range (*IQR*), and whiskers extend to $1.5 \times IQR$. Line represents the median. Distribution of values outside whiskers can be assessed in the contiguous density plots.

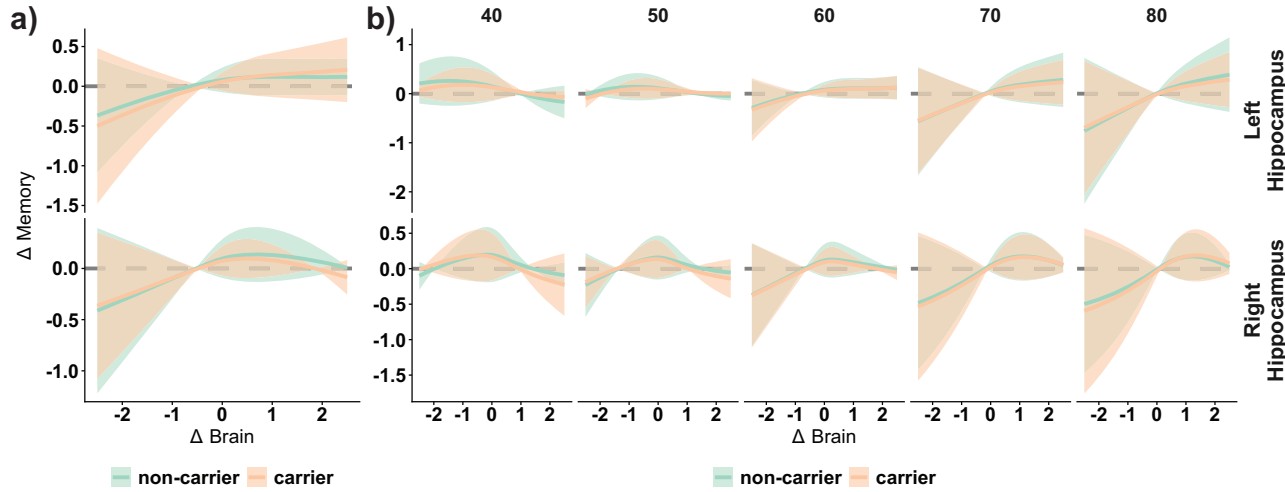

**Fig. 6 | APOE ε4 effect on brain change – memory change associations. a** Δbrain - Δmemory associations as a function of APOE ε4 (carriers vs. non-carriers) for the left and right hippocampus ($\beta_{wl} = -0.011$, edf = 1, $p_{unc} = .44$, $n = 3296$; $\beta_{wr} = -0.001$, edf = 1, $p_{unc} = .61$, $n = 3293$). **b** Δbrain - Δmemory associations as a function of APOE ε4 (carriers vs. non-carriers) and age for the left and right hippocampus. None of the terms were significant ($\beta_{wl} = .044$, edf = 1, $p_{unc} = .66$, $n = 3296$; $\beta_{wr} = -0.028$, edf = 1, $p_{unc} = .66$, $n = 3293$). Black line and ribbons represent mean and 95% CIs. P-values were obtained via GAMM-bootstrapping. P-values were obtained via wild bootstrapping. $\beta_w$ = Density-weighted betas, using the derivative function.

distribution with mean decline indicative of long-term, degenerative changes. This component is universal, as most individuals exhibited some degree of decline, with negative skewness arising from a subset of individuals undergoing accelerated brain aging[16,20,22,32,61]. The second source was modeled as a Gaussian distribution centered around zero, representing measurement error and other short-term influences[20,51,52]. Memory decline was modeled as linearly related to the brain aging component, plus random Gaussian noise. To explore the moderating effects (or lack thereof) of age and APOE ε4, we adjusted the parameters of the brain aging component, including mean, dispersion (e.g., variability), and skewness.

The simulation results revealed that observed brain decline was non-linearly associated with memory decline, with the relationship flattening among brain maintainers (Fig. 7a) mimicking the empirical change–change associations. Increasing the variance of the brain aging component strengthened the change–change associations and affected brain maintainers, replicating the moderating effect of age (Fig. 7c). Conversely, increasing mean decline and skewness did not alter the change–change associations despite leading to steeper mean decline in both observed brain and memory measures (Fig. 7b, d). See details in SI.

## Discussion

By mega-analyzing data from over 3700 cognitively healthy adults and 13 independent longitudinal studies, we found that changes in brain structure are associated with changes in episodic memory across

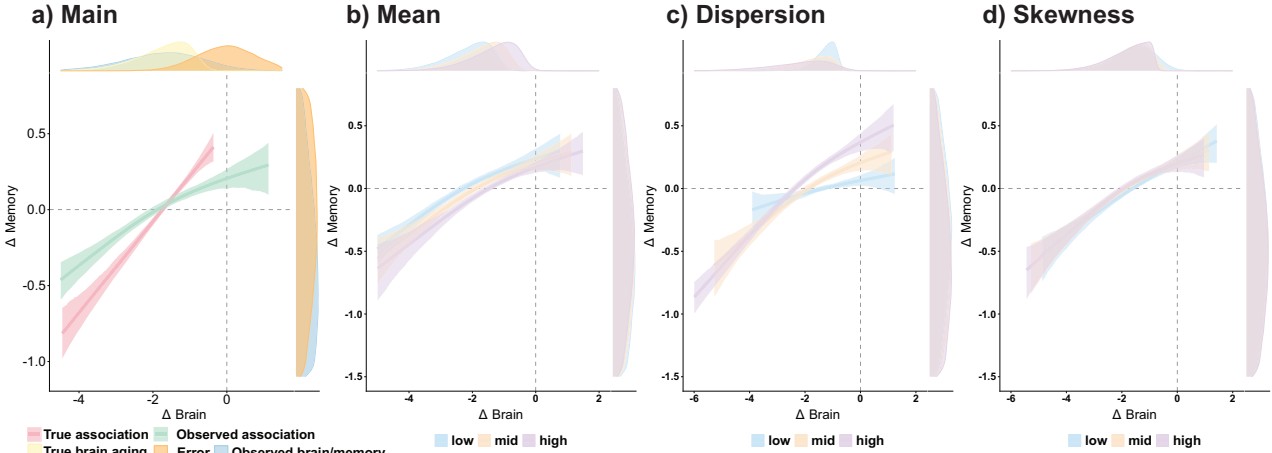

**Fig. 7 | Theoretical basis for brain change – memory change associations. a** The main simulations demonstrate how a skewed true (latent) distribution of brain decline, which has a linear association with memory decline (red line), results in non-linear observed associations between observed (measured) brain decline and memory decline (green line) due to the measurement noise. Density plots illustrate different distributions: yellow represents true brain and memory decline, orange represents measurement error, and blue represents observed brain and memory.

**b–d** The impact of distribution moments on true brain distribution on observed Δbrain – Δmemory associations. Specifically, the effect of **b** mean decline, **c** dispersion, and **d** negative skewness. Density plots correspond to the underlying distribution of true brain and memory decline. Black lines and ribbons indicate mean and 95% CIs across N = 1000 simulations. GAMs were used to fit the associations. See SI for more details.

several cortical and subcortical areas, with the strongest associations in the MTL. These associations became more pronounced with increasing age, while no evidence was found for stronger relationships in APOE ε4 carriers. We argue for common macrostructural systems supporting memory function, where multiple factors converge to increase vulnerability in older age. The implications of these findings are discussed below.

### Brain decline - memory loss associations: a generalized phenomenon or constrained to above-average brain decliners?

Observed brain decline is non-linearly associated with memory decline, with stronger associations in brain decliners, i.e., individuals exhibiting above-average brain decline relative to their age and sex, compared to brain maintainers. This finding within the context of cognitively healthy aging differs from previous research, which may have over-relied on linear regression models and modest sample sizes. At first sight, it suggests change – change associations are constrained to a specific population of individuals with steeper brain decline, at-risk for pathological neurodegeneration. Yet, our simulations challenge this interpretation, rather suggesting the non-linear trends result from the presence of multiple sources contributing to noisy measures of brain change. Amongst these, one component, i.e., brain aging, has linear associations with memory decline and is characterized by a negative mean and skewness. These assumptions align well with current evidence, including, critically, the skewed distribution of brain aging[16,20,22,32,61,62]. The findings suggest that changes in brain aging are dimensional, skewed, and an inherent process, which is an important determinant of memory loss in cognitively unimpaired elderly. Rather than a categorical view, where a degenerative component of brain change is limited to vulnerable individuals, it is the combination of noise and a skewed distribution that limits our ability to observe empirical associations in individuals with less brain change. These results likely underpin other categorical distinctions in neurocognitive aging[12] and are closely aligned with brain maintenance theory predictions[63].

### Brain change–Memory Change Associations in APOE ε4 Carriers: Distinct Decline, Shared Mechanisms?

APOE ε4 is associated with steeper brain decline and memory decline but does not affect the change – change associations between brain structure and memory. Carrying the APOE ε4 allele is the strongest

genetic risk for sporadic AD, with dose-dependent effects[64,65]. Older carriers of the APOE ε4 allele exhibited steeper memory and brain decline, particularly in the hippocampi and the right amygdala, aligning with many other studies[45,46]. It is plausible that a higher proportion of APOE ε4 carriers are on a path to clinical disease manifestation of AD, putatively driven by the spreading of Tau deposition, which is strongly linked to steeper brain atrophy, memory decline, and short-term clinical diagnosis[66]. However, the change – change associations and the moderating effects of age were not influenced by APOE ε4 status, nor are they likely affected by preclinical AD. First, in some regions, the change – change associations were evident before age 60, when the prevalence of Tau deposition is generally very low[67]. Aβ deposition before age 60 is slightly more common, but when controlling for Tau, the influence of Aβ on brain and memory decline is modest at best[66,68–70]. Second, the associations are not constrained to the MTL, where earlier preclinical changes are observed in AD. Third, if preclinical AD were to influence the Δbrain - Δmemory associations, APOE ε4 non-carriers would display attenuated or age-delayed associations between brain and memory decline. One study using linear models reported stronger associations between hippocampal change and memory decline in cognitively healthy APOE ε4 carriers[9], arguing that APOE ε4 carriers had a more hippocampal-centric pattern of atrophy in line with categorical aging and disease models. The current results align with a more hippocampal-centric pattern of atrophy but also fit well with a dimensional view of aging, where APOE ε4 contributes to accelerated brain aging, without changing the macrostructural mechanisms underlying the change – change associations. This is captured by the simulation analyses, which illustrate that a brain aging distribution with either steeper mean decline or higher skewness in APOE ε4 carriers would lead to both steeper brain and memory decline, but similar strength of the change – change associations. Altogether, these results fit with a dimensional view of aging, where APOE ε4 and early preclinical change in AD are one of many pathways affecting common biological substrates that determine memory function in older age, namely, regional and global macrostructural atrophy.

### Age is an important determinant of brain change – memory change associations

Age strengthens the associations between brain decline and memory decline, gradually extending to brain maintainers across most

significant regions. The hippocampi are among the earliest regions to exhibit these associations, emerging in the late fifties. This finding aligns with previous indirect evidence that the relationship between brain and memory decline strengthens with age[25,41,42]. Simulated data identified dispersion - greater variability in brain change across individuals - as the key factor driving stronger associations with age. This aligns with prior research indicating increased variability in both brain levels and brain change[16,20], a pattern also observed in cognition, including episodic memory[18,19]. Interestingly, mean decline did not significantly affect the strength of the brain - cognition associations. The steeper memory and MTL declines from around 60 years of age[10,11,15,16] are therefore not direct causes of these associations but rather serve as indirect markers. Since brain aging follows a unidirectional trajectory - where everyone experiences some degree of decline over time - greater variability in brain aging gives rise to steeper rates of brain decline. Overall, age is the primary determinant of degenerative brain change, and henceforth of change – change associations. Below a certain age, brain aging – or better said, population-level variability – is minimal, making it unlikely to be a key factor behind episodic memory loss; if a meaningful decline in episodic memory occurs in young adulthood[1,2,71]. What makes age the prime risk factor for brain decline and which age-related factors may explain variations in brain (and memory) decline, remain amongst the most critical questions in the field. All points to a multidimensional view, where brain systems, even in the absence of overt pathological changes, are highly vulnerable to several aging factors[32].

### Regional associations between brain change and memory change

Hippocampal atrophy unsurprisingly showed the strongest associations with memory decline over time. This is consistent with earlier studies in the context of cognitively healthy elderly[25–29], the key role of hippocampus in episodic memory[33,37], and its vulnerability to aging[32]. In contrast, ventricular associations likely reflect global, non-specific patterns of brain atrophy and have also been reported elsewhere[27,34]. The mechanisms underlying observed caudate, thalamus, putamen, and amygdala change – change associations remain unclear and require experimental approaches to move beyond speculation. One possibility is altered coupling between these regions and the hippocampus, as they all exhibit connectivity changes with the hippocampi during aging and episodic memory tasks[72–75]. Associations between cortical thinning and memory decline were weaker than those observed for subcortical structures[29]. Cortical thinning was chosen due to its high sensitivity to change; however, this sensitivity may render it more susceptible to influences unrelated to aging and long-term memory decline. Among the regions surviving multiple comparisons correction, the left parahippocampal gyrus stood out. Its anterior portion encompasses the entorhinal cortex, which serves as the main interface between the neocortex and the hippocampus, and is critically involved in memory[37,76,77]. Six additional regions in the temporal lobe were associated with memory decline, likely reflecting their roles in auditory, visual, or multimodal processing and integration. Most of the remaining regions, such as the pericallosal sulcus, the supramarginal gyrus, and the long insular gyrus, pertain to an action-mode network involved in task-positive, goal-directed behavior[78]. The overall pattern consists of relatively higher-order regions associated with both goal-directed and internal self-referential processing, which manifest the particular requirements that memory encoding and retrieval impose on the brain, that is, orchestrating dynamics amongst – often antagonistic - large-scale networks[79].

### Dimensionality of brain change: Global decline or regional contributions?

Previous research has suggested that the associations between brain atrophy and memory decline are driven either by a global factor of brain decline or are constrained to the MTL[6,25,39]. Our findings partially support both views, as we found evidence for a global factor of brain decline, while hippocampal atrophy showed the strongest regional associations with memory decline. However, the results reveal a more nuanced picture, with evidence for clustering observed. While the clustering solution made both topological and functional sense, caution is warranted regarding the specific solution obtained, as multiple solutions were plausible, and the input data were selected based on the somewhat arbitrary criterion of statistical significance. Nonetheless, several clustering solutions outperformed the one-factor solution, with some of these remaining associated with memory decline even after controlling for hippocampus or global decline. Most existing research, including this study, does not fully disentangle cross-regional correlations in brain aging from correlated errors. In any case, the current results indicate that a decline in regions critical to lower-order functions, such as attention, or indirectly related to memory via reward or executive control systems, contributes to memory loss independently of the integrity of medial lobe structures. Note also that the relationship between some observed regions and memory decline may be explained by a change in global cognition[39].

### Technical considerations and limitations

The study required considerable analytical flexibility, which may influence the outcome. A multiverse approach[80] was impractical due to constraints in data availability and computational resources. Applying iterative normative modeling and generating bootstrapped p-values across 166 regions would have rendered such an approach prohibitively time-consuming. Key considerations include: I) Normative modeling-based normalization using Hierarchical Bayesian Regression[81,82], a flexible technique that often outperforms other harmonization methods[83]. It standardizes Z-scores based on age and sex, making data relative and somewhat challenging to interpret. However, it eliminates most age-related homoscedasticity and facilitates comparisons with other research, since data are aligned to an openly available norm. Assuming similar scanner harmonization performance, normative models should yield results comparable to other methods except in cases where interindividual variability strongly depends on age or sex. This is generally not the case for cortical thickness and subcortical volumes, except for ventricular features that show substantial age-related changes in variability. Consequently, we expect some attenuation in brain-memory change associations for ventricles when using normative modeling. In contrast, estimates for the age × brain atrophy interaction should be largely method-independent and reflect standardized associations. II) Bootstrapped p-values were estimated to better control the false positive rate in GAMMs[84], leading to somewhat reduced power compared to linear models when the observed relationship is linear, although it can be argued that it hardly ever is. III) Estimation of change scores via linear changes over time per individual, with weighting applied to control for differences in longitudinal reliability as individuals with longer follow-up times and more observations contribute with more reliable data[20,85]. This represents a compromise choice, balancing data quality, flexibility, and interpretability; yet other approaches based, e.g., in standard equation modeling or use of random slopes as measures of individual change have also its strengths. IV) Inclusion of covariates representing other variables (e.g., neurochemical measurements) may account for unexplained variance and uncover further associations. V) Atlas choice was guided by practical considerations, particularly the availability of high-quality normative models[82]. While it is unclear which atlas would be optimal for our study, a multi-modal atlas may be a promising candidate[86]. In any case, we conducted a series of control analyses that included additional covariates, analytical choices, and age-restricted subsamples. These analyses consistently produced similar patterns of association, supporting the robustness of our main findings.

Here, we combined datasets to increase sample size and statistical power. However, we also inherited the idiosyncrasies of these datasets, such as inclusion/exclusion criteria, sample unrepresentativeness, and recruitment methods. Also, analytical compromises were made to ensure compatibility across all datasets. Memory function was harmonized independently within each dataset, and thus reflects the specific tests used rather than a common construct; most of these tests primarily assessed verbal episodic learning and free recall. While modality or content-specific analyses could, in principle, reveal more precise specific anatomical relationships, the available data did not permit such detailed investigation. Also, an item-response framework is theoretically a more robust approach for harmonisation, placing all individuals in the same space; yet, it is unfeasible in practice, as some datasets lacked shared tests. Estimating non-linear trajectories within individuals was inadvisable given the relatively limited number of observations and follow-up durations per participant. Reliable estimation of non-linear change typically requires substantially more data than linear models. Moreover, our preprocessing pipeline explicitly accounts for and removes non-linear age trends at the group level, thereby addressing a major source of non-linearity in brain and cognitive measurements.

The two-source model proposed in the discussion remains speculative, while the specific mechanisms underlying brain aging remain elusive. Our post-hoc simulation requires external empirical validation using more precise estimates of change than those available in current legacy datasets[87]. Improved study design and advanced scanning protocols, such as cluster scanning[88], offer a promising avenue to enable such validation. Young APOE ε4 carriers showed less memory change. We remain cautious of these results as the youngest segment of our sample had limited availability of APOE information, and thus emphasize the need for further research. For communication purposes, we used the terms brain maintainer and brain decliner despite employing a dimensional approach. This choice favors clarity over strict precision. For a more precise representation, readers are referred to the Supporting app, where point estimates are provided across varying levels of brain atrophy.

Overall, brain atrophy accounts for only a small proportion of the variance in memory decline, unsurprising given the substantial heterogeneity influencing interindividual differences in cognitive aging[89,90]. Other brain structure modalities, such as cortical area, are both associated with memory decline and independent from cortical thinning[29,35]. White matter integrity[91], particularly within limbic pathways[92], along with white matter hyperintensities[93] and other vascular factors, are also likely to contribute, at least partially independently. Additional potential contributors include brainstem, basal ganglia, and cortical neurochemistry[94,95] and indices of brain function[96]. Associations between memory decline and early pathological changes in limbic-predominant age-related TDP-43 encephalopathy (LATE-NC), primary age-related tauopathy (PART), or AD may be largely mediated by shared variance with temporal brain atrophy[97–99]. Even accounting for these factors, substantial variance in cognitive decline will remain unexplained due to measurement error, technological limitations, and resilience factors. The relationships among these factors, as well as our findings, are shaped by sample characteristics; accordingly, our results are most generalizable to other samples of clinically unimpaired individuals. Following up individuals after clinical conversion may produce modest quantitative shifts - such as stronger effects of brain atrophy, age × brain atrophy, and APOE ε4 on memory decline – consistent with established links among these measures and cognitive impairment. In contrast, qualitative changes, such as a reconfiguration of the regional pattern or its interaction with APOE ε4, are unlikely unless the sample is substantially clinically enriched. Only with significant pathology would a fundamental shift in the neurobiological basis of memory decline be expected. Multimodal integration of longitudinal neuroimaging and cognitive data with postmortem neuropathology offers a promising avenue for advancing our understanding[100].

To conclude, regional brain decline over time, particularly, but not limited to, the hippocampus, was associated with memory loss with age, but APOE ε4 status was not a key factor behind these associations. These associations strengthened with age from around 60 years. Methodologically, the findings underscore the necessity for methods and approaches that capture non-linear dynamics and put focus on the variability across individuals rather than mean change. Theoretically, the results support a multidimensional view of memory, aging, and disease, where multiple factors converge to increase the vulnerability of common macrostructural systems supporting memory function in older age.

## Methods
### Participants
The research complies with all relevant ethical regulations, and all participants provided informed consent. The main project was approved by the Norwegian Regional Committee for Medical Research Ethics South (REK sør-øst, approval no. 8122), and each dataset was approved by the relevant ethical review board, as specified in Supplementary Table 7.

In this study, we combined 13 ongoing or retrospective datasets that included a) cognitively healthy adult individuals with longitudinal assessments of brain structure (T1-weighted [T1w] sequence) and of memory function. All the main analyses were carried out using only longitudinal information from both brain structure and memory function. Individuals with only 1 observation, or uncoupled memory – brain data, were used only in preprocessing stages: for calibration purposes in MRI preprocessing and for principal component extraction, and Z-scoring of memory scores (see below). Including these individuals alongside those with follow-up data allowed for a more robust and generalizable estimation of sex-specific age trajectories - less influenced by attrition bias – since participants with follow-up data tend to be healthier than their age peers. This approach also enabled more accurate estimation of test-retest effects in memory assessments. Sex was determined by self-report. See Supplementary Tables 5 and 6 for information on the initial MRI and memory samples. Unless otherwise stated, we focus on the longitudinal-coupled samples used in the main analyses.

A total of 3737 cognitively healthy adults, with at least partially overlapping longitudinal follow-ups of brain structure and memory function, with a minimum total span of 1.5 years, were included in the analyses. In total, 10,343 MRI observations and 13,460 memory observations contributed to the analyses (Table 1, Supplementary Fig. 1). The datasets include the LCBC[101], Betula[102], UB[103,104], and BASE-II[105,106] datasets (from the Lifebrain Consortium)[107] as well as the COGNORM[108], the Alzheimer's Disease Neuroimaging Initiative (ADNI) database (https://adni.loni.usc.edu)[109], AIBL[110], BBHI[111], the Harvard Aging Brain Study (HABS)[112], the UKB[113], PREVENT-AD[114,115], OASIS3[116], and VETSA[117] datasets. In addition to cohort-specific inclusion and exclusion criteria, observations concurrent with cognitive impairment and Alzheimer's dementia were excluded. Individuals with baseline age <18 years, or with severe neurological or psychiatric disorders, were additionally excluded. Based on preprocessing requirements, MRI data from scanners with fewer than 25 observations were excluded, as well as individuals with less than 1.5 years of follow-up of either memory function or brain structure. Individuals without partially overlapping follow-up periods of brain and memory assessment were excluded, as well as those with non-overlapping periods >10 years. See Supplementary Table 7 for data availability, ethical standards, and contact information and SI for more sample details. A total of 3,149 subjects had APOE data available. Of these, 27.8 % were carriers of the APOE ε4 allele.

## Memory function

For each sample, we first z-normalized all measures based on the first time point and the different available memory tests. When multiple measures were available, we estimated a main component using Principal Component Analysis (PCA; *prcomp*) with all measures at the first time point as inputs. Missing values were imputed using *imputePCA* from the *missMDA* r-package[118]. Only for OASIS3, the imputed number of values was not negligible (>.5%). See Supplementary Table 8 for information on memory data for each sample. For each dataset, we regressed out age as a smoothing term, sex, and one or two dummy test-retest regressors using GAMMs (*mgcv R-package*)[119]. Individual identifiers were used as random intercepts, and the number of dummy test-retest regressors depended on whether the dataset had 2 or >=3 waves with memory function data. We retained individuals with at least two observations and a minimum follow-up of 1.5 years. For each individual, we then estimated yearly change by regressing memory observations on follow-up time, which were Z-standardized by site and fed to higher-level analyses.

## MRI preprocessing and brain structure

**MRI acquisition and preprocessing.** Structural T1w MPRAGE and FSPGR scans were collected using 1.5 and 3 T MRI scanners. See information on scanner parameters and scanners across datasets in Supplementary Table 9. Data was converted to BIDS[120] and pre-processed using the longitudinal FreeSurfer v.7.1.0 stream[121] for cortical reconstruction and volumetric segmentation of the structural T1w scans[122,123]. See details in SI. Data was tabulated based on the *Destrieux* (cortical; N = 74 regions per hemisphere)[56] and *aseg* (subcortical, N = 18 regions) atlases[57]. Atlas choice was guided by practical considerations, as high-quality normative models were readily available[82].

**Data harmonization.** Brain regions were harmonized using a normative modeling framework, resulting in site-agnostic deviation scores (z-scores) adjusted for age and sex[61,82] based on a Hierarchical Bayesian Regression technique[81] as implemented in the *PCNtoolkit* (v.0.30.post2), in *Python3* environment[124] (v.3.9.5). Calibration to the model was performed iteratively (N = 100) to avoid losing longitudinal observations. This step was carried out with the initial MRI sample, i.e., regardless of the availability of longitudinal MRI data or paired memory function assessments. Calibrated data, across iterations, showed high reliability. See SI for more normative modeling harmonization details. Next, we selected individuals with at least two observations and a minimum follow-up of 1.5 years. For each individual and region, we estimated yearly change by regressing normative MRI values on follow-up time, Z-standardized data by site, and fed the output into higher-level analyses. The correlation between individual change estimates derived from the normative-modeling approach and those obtained using a GAMM – with sex as fixed factor, age as smoothing term, and subject, scanner, and dataset as random intercepts[58] – was very high across most regions, suggesting a null-to-modest impact of this analytical choice (mean r = .96 across regions; see Supplementary Fig. 6).

## Higher-level analyses

All the analyses were carried out in the R environment (v.4.2.1)[125]. Visualizations were made with the *ggplot2* (v.4.0.0)[126] and the *ggseg* (brain images; v.1.6.5)[127] R-packages. Regions were selected among the statistically significant results to best illustrate the range of change-change associations. Visualizations for all regions are available in the Supporting app. Analyses were mostly carried out using *gamm* models as implemented in the *mgcv* (v.1.9-3) *R-package*[119]. Derivatives were estimated based on finite differences as implemented in the *gratia* package (v.0.11-1)[128]. Linear mixed models as implemented in *lme4* (v.1.1-37), *lmerTest* (v.3.1-3)[129,130] were also used to assess the effect of APOE ε4 on brain and cognition.

To test the regional association between brain change and memory change, we carried out univariate weighted GAMMs, with a smooth term of Δbrain predicting Δmemory. An Δbrain × age tensor interaction term was added, as well as a smooth term of age, to assess the effect of age on Δbrain – Δmemory associations. The effect of APOE ε4 on memory and brain regions was tested using weighted linear mixed effects models, with APOE ε4 status predicting either Δbrain or Δmemory. The age × APOE ε4 status interaction was tested using GAMMs with age as a smooth term by APOE ε4 status as an ordered factor, in addition to APOE ε4 status as a fixed effect, and age as a smooth term. This effectively models the smooth term of age for APOE ε4 non-carriers as a reference, while the smoothed term for APOE ε4 carriers models the difference with respect to the reference. Similarly, the Δbrain × APOE ε4 status was used to assess the effect of APOE ε4 on Δbrain – Δmemory associations. Finally, age × Δbrain × APOE ε4 status with their simple effects) was used to test a triple interaction of age, APOE ε4 status, and Δbrain on Δmemory. All models included the dataset as a random intercept.

We tested the dimensionality by performing PCA and clustering on the regions (N = 19) showing significant Δbrain – Δmemory associations. To ensure the resulting global atrophy factor was aligned with the association of interest, we constrained it to regions linked to memory decline. As such, it may be more appropriate to interpret this factor as reflecting the common contribution of memory-sensitive areas. Therefore, the global factor identified here may not be fully generalizable to broader measures of global brain atrophy. This, along with the heterogeneity introduced by combining multiple cohorts, may explain the relatively modest variance explained by the global factor compared to those reported elsewhere[6,29]. The clustering was based on the M3C, Monte-Carlo Reference-based Consensus algorithm, implemented in the M3C (v.1.30.0) package[131], which, critically, tests whether the desired solution is better than K = 1. We used a spectral clustering algorithm and PAC criteria, while the remaining parameters were set to the default. As post-hoc analyses, we tested whether the resulting clusters of brain change were related to memory change, controlling for the effect on memory of other brain regions such as the hippocampus or a general factor of brain decline using GAMMs as described above. See more details on SI along with pseudocode.

Note that in all analyses we have one observation per individual (e.g., Δmemory) as we are using change scores. Note also that age (and sex) trends are removed, and thus the model captures only inter-individual associations - relative change - and age trends are uninterpretable. Prior to any analysis, outlier values, defined as values > 4.5 SD from the mean, were removed from the analyses (based on a $p < 0.05$ of observing at least one outlier value given a normal distribution and our sample size). In GAMMs, we estimated p-values using a wild bootstrapping (n = 5000) as the out-of-the-box p-values, as implemented in *mgcv*, are anticonservative[84]. Wild bootstrapping generates a null distribution of p-values by a) estimating a null model without the regressor of interest, b) extracting predicted values from the model and its residuals, c) adding the predicted value to the residuals multiplied by a random vector of 1 and -1s, and d) re-estimating a new model using this score as the predicted variable. When appropriate, p-values were corrected for multiple comparisons using FDR[132]. All models used weights to account for unequal reliability of longitudinal data. That is, individuals with short follow-up periods and fewer observations contribute with more unreliable, high-variance data and thus should produce an unequal spread of residuals. We used the square of reliability as weights, as estimated elsewhere[20]. Weights <.09, corresponding to longitudinal reliability <.3, were set at 0.09. For tensor interactions, we estimated the derivatives along Δbrain at specific ages (40, 50, 60, 70, 80 years) using a finite differences approach. The degree of association between Δbrain and Δmemory is estimated

only for Δbrain <0 – as most associations are constrained in brain decliners, estimating the mean association across the Δbrain weighted by the density of data-points. Note that the β coefficients are standardized and, therefore, somewhat comparable to an effect size measure. We estimated the partial variance explained by brain change and by brain change × age interaction as the difference in variance explained between the full model and a null model without the regressors of interest. This approximation likely underestimates the variance explained by the regressors as i) mgcv's default variance explained metric penalizes model complexity, ii) smooth terms are re-fitted in the null model, and iii) variance attributable to the regressors may be partially absorbed by re-fitting the random-effect components. Data from the ventricles were sign-reversed. We slightly trimmed the x-axis in the figures – removing ≈1% of the observations – to exclude high uncertainty fittings from visualization.

### Reporting summary

Further information on research design is available in the Nature Portfolio Reporting Summary linked to this article.

## Data availability

The raw data were gathered from 13 different datasets. Different agreements are required for each dataset. Most datasets are openly available with prespecified data usage agreements. For some datasets, such as UKB, fees may apply. Requests for Lifebrain cohorts (LCBC, Umeå, UB) and COGNORM, should be submitted to the corresponding principal investigator. See data availability and contact details for all datasets in Supplementary Table 7. Group-level estimates generated in this study are provided in the Supporting app (https://vidalpineiro.shinyapps.io/brain_mem_change/). Source data to reproduce the figures is provided in a separate Source Data file. Source data are provided with this paper.

## Code availability

Statistical analyses in this manuscript are available at https://github.com/daidak/memory-brain-change [133]. All analyses were performed in R. The scripts were run on the Colossus processing cluster, University of Oslo. MRI preprocessing and feature generation scripts were performed with FreeSurfer (https://surfer.nmr.mgh.harvard.edu/) software.

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

## Acknowledgements

This work was supported by the Department of Psychology, University of Oslo (to K.B.W., A.M.F.), the Norwegian Research Council (to K.B.W. [325001, 301395, 239889], A.M.F. [325878, 262453], D.V.P [324882]), the project has received funding from the European Research Council's Starting Grant scheme under grant agreements 283634, 725025 (to A.M.F.) and 313440 (to K.B.W.), and the University of Oslo through the UiO:Life Science convergence environment [AHeadForLife: Societal and environmental determinants of brain and cognition] (to A.M.F). R.N.H. was supported by the UK Medical Research Council [SUAG/046/G101400]. A.P.-L. was partly supported by grants from the National Institutes of Health (RO1AG076708), Jack Satter Foundation, and BrightFocus Foundation. G.C. and J.S-S. were partially supported by the Spanish Ministry of Science and Innovation (PID-2022-139298OA-C22). This work was supported by a grant from the Simons Foundation (AN-NC-AB-Research-01169072 to A.M.F., K.B.W., L.N.). The different sub-studies are supported by different sources. LCBC: the Norwegian Research Council (to A.M.F., K.B.W.), and the National Association for Public Health's dementia research program (A.M.F.). Umeå (betula): a scholar grant from the Knut and Alice Wallenberg (KAW) foundation to L.N. UB: D.B.F. was funded by the ICREA Academia Award (2019) and the 2014 awards from the Catalan Government. He acknowledges the CERCA Programme/Generalitat de Catalunya and is supported by the María de Maeztu Unit of Excellence (Institute of Neurosciences, University of Barcelona) MDM-2017-0729, and a PID2022-137234OB-100 projects, funded by the Ministry of Science, Innovation and Universities and ERDF/EU. BASE-II. BASE-II has been supported by the German Federal Ministry of Education and Research under grant nos 16SV5537, 16SV5837, 16SV5538, 16SV5536K, 01UW0808, 01UW0706, 01GL1716A, and 01GL1716B, and by the European Research Council under grant agreement no. 677804 (to S.K.). BBHI. The data from BBHI was obtained with funding from "la Caixa" Foundation (grant agreement n° LCF/PR/PR16/11110004), and also from Institut Guttmann and Fundació Abertis. COGNORM is funded by the South-Eastern Norway Regional Health Authorities (#2017095). The Norwegian Health Association (#19536) and by Wellcome Leap's Dynamic Resilience Program (jointly funded by Temasek Trust) #104617). The funding sources had no role in the study design. Data used in preparation of this article were obtained from the Alzheimer's Disease Neuroimaging Initiative (ADNI) database (adni.loni.usc.edu). The ADNI was launched in 2003 as a public-private partnership, led by Principal Investigator Michael W. Weiner, MD. The primary goal of ADNI has been to test whether serial magnetic resonance imaging (MRI), positron emission tomography (PET), other biological markers, and clinical and neuropsychological assessment can be combined to measure the progression of mild cognitive impairment (MCI) and early Alzheimer's disease (AD). For up-to-date information, see https://adni.loni.usc.edu/. As such, the investigators within the ADNI contributed to the design and implementation of ADNI and/or provided data but did not participate in analysis or writing of this report. A complete listing of ADNI investigators can be found at: http://adni.loni.usc.edu/wp-content/uploads/how_to_apply/ADNI_Acknowledgement_List.pdf. Data collection and sharing for this project were funded by the ADNI (NIH Grant U01 AG024904). ADNI is funded by the National Institute on Aging, the National Institute of Biomedical Imaging and Bioengineering, and through generous contributions from the following: AbbVie, Alzheimer's Association; Alzheimer's Drug Discovery Foundation; Araclon Biotech; BioClinica, Inc.; Biogen; Bristol-Myers Squibb Company; CereSpir, Inc.; Cogstate Eisai Inc.; Elan Pharmaceuticals, Inc.; Eli Lilly and Company; EuroImmun; F. Hoffmann-La Roche Ltd and its affiliated company Genentech, Inc.; Fujirebio; GE Healthcare; IXICO Ltd.; Janssen Alzheimer Immunotherapy Research & Development, LLC.; Johnson & Johnson Pharmaceutical Research & Development LLC.; Lumosity; Lundbeck; Merck & Co., Inc.; Meso Scale Diagnostics, LLC.; NeuroRx Research; Neurotrack Technologies; Novartis Pharmaceuticals Corporation; Pfizer Inc.; Piramal Imaging; Servier; Takeda Pharmaceutical Company; and Transition Therapeutics. The Canadian Institutes of Health Research is providing funds to support ADNI clinical sites in Canada. Private sector contributions are facilitated by the Foundation for the National Institutes of Health (http://www.fnih.org). The grantee organization is the Northern California Institute for Research and Education, and the study is coordinated by the Alzheimer's Therapeutic Research Institute at the University of Southern California. ADNI data are disseminated by the Laboratory for Neuro Imaging at the University of Southern California. Data used in the preparation of this article was obtained from the Australian Imaging Biomarkers and Lifestyle flagship study of ageing (AIBL) funded by the Commonwealth Scientific and Industrial Research Organisation (CSIRO) which was made available at the ADNI database (www.loni.usc.edu/ADNI). The AIBL researchers contributed data but did not participate in analysis or writing of this report. AIBL researchers are listed at www.aibl.csiro.au. Parts of the data used in the preparation of this article were obtained from the Harvard Aging Brain Study (HABS - P01AG036694; https://habs.mgh.harvard.edu). The HABS study was launched in 2010, funded by the National Institute on Aging. and is led by principal investigators Reisa A. Sperling,

MD, and Keith A. Johnson, MD at Massachusetts General Hospital/Harvard Medical School in Boston, MA." OASIS data were provided [in part] by OASIS 3: Longitudinal Multimodal Neuroimaging: Principal Investigators: T. Benzinger, D. Marcus, J. Morris; NIH P30 AG066444, P50 AG00561, P30 NS09857781, P01 AG026276, P01 AG003991, R01 AG043434, UL1 TR000448, R01 EB009352. AV-45 doses were provided by Avid Radiopharmaceuticals, a wholly owned subsidiary of Eli Lilly. PREVENT-AD was funded by the Canadian Institutes of Health Research, McGill University, the Fonds de Recherche du Québec – Santé, Alzheimer's Association, Brain Canada, the Government of Canada, the Canada Fund for Innovation, the Douglas Hospital Research Centre and Foundation, the Levesque Foundation, an unrestricted research grant from Pfizer Canada. Private sector contributions are facilitated by the Development Office of the McGill University Faculty of Medicine and by the Douglas Hospital Research Centre Foundation (http://www.douglas.qc.ca/). UK Biobank is generously supported by its founding funders the Wellcome Trust and UK Medical Research Council, as well as the Department of Health, Scottish Government, the Northwest Regional Development Agency, British Heart Foundation and Cancer Research UK. The organisation has over 150 dedicated members of staff, based in multiple locations across the UK. VETSA Parts of the data are from VETSA, which is funded by the National Institute of Aging grants R01s AG018384, AG018386, AG050595, AG022381, AG076838. The content is the responsibility of the authors and does not necessarily represent official views of the NIA, NIH, or VA. U.S. Department of Veterans Affairs, Department of Defense; National Personnel Records Center, National Archives and Records Administration; Internal Revenue Service; National Opinion Research Center; National Research Council, National Academy of Sciences; and the Institute for Survey Research, Temple University provided invaluable assistance in the conduct of the VET Registry. The Cooperative Studies Program of the U.S. Department of Veterans Affairs provided financial support for development and maintenance of the Vietnam Era Twin Registry. We would also like to acknowledge the continued cooperation and participation of the members of the VET Registry and their families.

## Author contributions

D.V.P. Conceptualization, Methodology, Formal analysis, Writing - Original Draft; Ø.S. Methodology, Software, Formal analysis, Writing - Review & Editing; M.S. Resources, Data Curation, Writing - Review & Editing; I.K.A. Software, Writing - Review & Editing; W.B. Writing - Review & Editing; D.B-F. Resources, Writing - Review & Editing; A.B. Writing - Review & Editing; G.C. Resources, Writing - Review & Editing; S.D. Resources, Data Curation, Writing - Review & Editing; P.G. Methodology, Writing - Review & Editing; R.N.H. Conceptualization, Writing - Review & Editing; S.K. Writing - Review & Editing; U.L. Resources, Writing - Review & Editing; A.M.M. Software, Writing - Review & Editing; L.N. Conceptualization, Resources, Writing - Review & Editing; A.P.L. Conceptualization, Resources, Writing - Review & Editing; J.M.R. Data Curation, Writing - Review & Editing; J.S-S. Resources, Writing - Review & Editing; C.S.P. Resources, Writing - Review & Editing; L.O.W. Resources, Writing - Review & Editing; T.W. Methodology, Software, Formal analysis, Writing - Review & Editing; K.B.W. Conceptualization, Resources, Writing - Review & Editing; A.M.F. Conceptualization, Resources, Writing, Review & Editing.

## Competing interests

A.P.-L. serves as a paid member of the scientific advisory boards for Neuroelectrics, Magstim Inc., TetraNeuron, Skin2Neuron, MedRhythms, and AscenZion. He is the co-founder of TI Solutions and the co-founder and chief medical officer of Linus Health. A.P.-L. is listed as an inventor on several issued and pending patents on the real-time integration of transcranial magnetic stimulation with electroencephalography and magnetic resonance imaging, and applications of noninvasive brain stimulation in various neurological disorders; as well as digital biomarkers of cognition and digital assessments for early diagnosis of dementia. The remaining authors declare no competing interests.

## Additional information

Didac Vidal-Piñeiro [1] ✉, Øystein Sørensen [1], Marie Strømstad[1], Inge K. Amlien [1], William Baaré [2], David Bartrés-Faz [3,4,5], Andreas M. Brandmaier [6,7,8,9], Gabriele Cattaneo[4,10], Sandra Düzel[6], Paolo Ghisletta [11], Richard N. Henson [12], Simone Kühn [6,9,13,14], Ulman Lindenberger[6,8,9], Athanasia M. Mowinckel [1], Lars Nyberg [1,15,16,17], Alvaro Pascual-Leone [18,19], James M. Roe[1], Javier Solana-Sánchez[4,10], Cristina Solé-Padullés[3,5], Leiv Otto Watne [20,21], Thomas Wolfers[22], for the Vietnam Era Twin Study of Aging (VETSA)*, Kristine B. Walhovd [1,23] & Anders M. Fjell[1,23]

[1]Centre for Lifespan Changes in Brain and Cognition, Department of Psychology, University of Oslo, Oslo, Norway. [2]Danish Research Centre for Magnetic Resonance, Department of Radiology and Nuclear Medicine, Copenhagen University Hospital-Amager and Hvidovre, Copenhagen, Denmark. [3]Department of

Medicine, Faculty of Medicine and Health Sciences, Institute of Neurosciences, University of Barcelona, Barcelona, Spain. [4]Institut Guttmann, Institut Universitari de Neurorehabilitació adscrit a la UAB, Badalona, Barcelona, Spain. [5]Institut d'Investigacions Biomèdiques August Pi i Sunyer (IDIBAPS), Barcelona, Spain. [6]Center for Lifespan Psychology, Max Planck Institute for Human Development, Berlin, Germany. [7]Department of Psychology, MSB Medical School Berlin, Berlin, Germany. [8]Centre for Computational Psychiatry and Ageing Research, Max Planck Institute for Human Development, Berlin, Germany. [9]Max Planck UCL Centre for Computational Psychiatry and Ageing Research, London, UK. [10]Fundació Institut d'Investigació en Ciències de la Salut Germans Trias i Pujol, Barcelona, Spain. [11]Faculty of Psychology and Educational Sciences, University of Geneva, Geneva, Switzerland. [12]MRC Cognition and Brain Sciences Unit, Department of Psychiatry, University of Cambridge, Cambridge, United Kingdom. [13]Department of Psychiatry and Psychotherapy, Department of Psychiatry, University Medical Center Hamburg-Eppendorf, Hamburg, Germany. [14]Center for Environmental Neuroscience, Max Planck Institute for Human Development, Berlin, Germany. [15]Umeå Center for Functional Brain Imaging, Umeå University, Umeå, Sweden. [16]Department of Medical and Translational Biology, Umeå University, Umeå, Sweden. [17]Department of Diagnostics and Intervention, Umeå University, Umeå, Sweden. [18]Hinda and Arthur Marcus Institute for Aging Research, Harvard Medical School, Boston, MA, USA. [19]Department of Neurology, Harvard Medical School, Boston, MA, USA. [20]Oslo Delirium Research Group, Institute of Clinical Medicine, Campus Ahus, University of Oslo, Oslo, Norway. [21]Department of Geriatric Medicine, Akershus University Hospital, Lørenskog, Norway. [22]Department of Psychiatry and Psychotherapy, German Center for Mental Health, University Clinic Tübingen, Tübingen, Germany. [23]Computational Radiology and Artificial Intelligence, Department of Radiology and Nuclear Medicine, Oslo University Hospital, Oslo, Norway. *A list of authors and their affiliations appears at the end of the paper. ✉e-mail: d.v.pineiro@psykologi.uio.no

## for the Vietnam Era Twin Study of Aging (VETSA)

Carol E. Franz[24], William S. Kremen[24], Michael J. Lyons[25], Anders M. Dale[24], Jeremy A. Elman[24], Christine Fennema-Notestine[24], Matthew S. Panizzon[24], Chandra A. Reynolds[26,27], Katherine Bangen[24], Tyler R. Bell[24], Corwin Boake[28], Divya S. Bolar[24], Randy Buckner[29], Alice Cronin-Golomb[25], Stephen M. Dorros[24], Lindon Eaves[30], Seth Eisen[31,32], Lisa T. Eyler[24], Nathan A. Gillespie[30], Eric Granholm[24], Daniel E. Gustavson[27], Donald J. Hagler Jr[24], Richard L. Hauger[24], Diane Jacobs[24], Kristen Jacobson[33], Amy Jak[24], Gig Levine[34,35], Thomas Liu[24], Mark Logue[25], Linda K. McEvoy[24], Ruth E. McKenzie[25,36], Sally Mendoza[35], Michael C. Neale[30], Robert A. Rissman[24], Nicholas Smith[31,37], Rongxiang Tang[38], Rosemary Toomey[25], Ming Tsuang[24], Xin M. Tu[24], Art Wingfield[25,39], Hong Xian[40], Dirk Hellhammer[41], Warner Schaie[31], Sonia Lupien[42], Bruce Fischl[29,43], Larry Seidman[29,44], Brinda Rana[24], Terry Jernigan[24], Claire Murphy[45], Wesley Thompson[24,46] & Nicholas Schork[24,47]

[24]University of California San Diego, La Jolla, CA, USA. [25]Boston University, Boston, MA, USA. [26]University of California Riverside, Riverside, CA, USA. [27]University of Colorado Boulder, Boulder, CO, USA. [28]Baylor College of Medicine, Houston, TX, USA. [29]Harvard University, Cambridge, MA, USA. [30]Virginia Commonwealth University, Richmond, VA, USA. [31]University of Washington, Seattle, WA, USA. [32]St. Louis Veterans Affairs, St. Louis, MO, USA. [33]University of Chicago, Chicago, IL, USA. [34]Stanford University, Stanford, CA, USA. [35]University of California Davis, Davis, CA, USA. [36]Merrimack college, North Andover, MA, USA. [37]Seattle Veteran Affairs, Seattle, WA, USA. [38]Texas A&M University, College Station, TX, USA. [39]Brandeis University, Waltham, MA, USA. [40]St. Louis University, St. Louis, MO, USA. [41]University of Trier, Trier, Germany. [42]University of Montreal, Montreal, QC, Canada. [43]Massachusetts General Hospital, Boston, MA, USA. [44]Massachussets Mental Health Center, Boston, MA, USA. [45]San Diego State University, San Diego, CA, USA. [46]Laureate Institute for Brain Research, Tulsa, OK, USA. [47]Translational Genomics Research Institute, Phoenix, AZ, USA.

