## [Transparent Peer Review file · Nature Communications]

Vulnerability to memory decline in aging revealed by a mega-analysis of structural brain change

Corresponding Author: Dr Didac Vidal

Version 0:

Reviewer comments:

Reviewer #1

(Remarks to the Author)

This study performed a mega-analysis of 13 longitudinal datasets, including 3737 cognitively healthy adults with available data on MRI and memory. The key research question was to examine change-change associations between brain structure and memory. The study confirmed an association between brain atrophy and memory decline, and concluded that it primarily affected individuals with above average brain atrophy. The authors further concluded that the associations were more pronounced in the hippocampus and strengthened with age. Carrying the APOE $\epsilon 4$ risk allele did not affect the change-change associations.

The manuscript is well written and the authors have done an impressive work combining data from multiple studies. The authors further did a careful job explaining what they did, motivating their decisions for the statistical analyses, and discussing strengths and limitations of their methods. The findings seem robust and the conclusions are supported by the results.

The study is an important addition to the literature given that these analyses combine large and diverse samples to address a universal and important research question. The results of the study are in line with previous smaller studies indicating that brain atrophy, particularly in the hippocampus, is a driving factor in memory decline in aging.

It would benefit the manuscript even further to provide some more detail on how to interpret the findings.

- How much of the variance in episodic memory change is explained by change in brain structure. Could the authors provide some further information on the effect size for the overall effect?
- Similarly, how big proportion of the association is explained by regional vs global brain atrophy?
- Could the authors elaborate a bit more on what other factors may underly episodic memory change (i.e. the variance that is not explained by brain atrophy)? Which future studies should be conducted in order to answer that question?
- Given that the study populations were restricted to cognitively unimpaired participants. Would the authors agree that their conclusions would mainly apply to normal aging and it is possible that the patterns would be different in a sample including also individuals who are close to a dementia diagnosis? Or do the authors think that the “brain decliners” are in fact individuals in close proximity to dementia?
- Especially regarding the conclusion regarding the $\epsilon 4$ carriers, would the authors expect the same pattern in a sample where many are expected to develop dementia in the near future?

Some specific points:

Introduction, line 178: what do the authors mean by the term “at-risk individuals”? They seem to be referring to individuals who decline faster than average but the term becomes circular.

Methods: the study investigated 166 cortical and subcortical brain regions. Why were so many regions included? How were they selected?

Reviewer #2

(Remarks to the Author)

The manuscript addresses important questions regarding associations between changes in brain structure and episodic memory across the lifespan, leveraging a robust mega-analytical dataset and sophisticated statistical methods. Longitudinal analyses are valuable for exploring nonlinear trajectories and disentangling the complex interrelationships among brain, memory, aging, and genetic factors. This study demonstrates considerable effort in integrating multisource datasets with extensive data cleaning, processing, and modeling. However, several major and minor issues warrant clarification, refinement, or additional analysis.

Major Comments

1. The manuscript proposes a dimensional view of aging and brain decline, yet categorically separates individuals into "above-average decliners" and "maintainers." This introduces theoretical inconsistency. The authors should reconcile these approaches and consider aligning their framework with cognitive reserve, compensation, or resilience theories.
2. In a similar vein, the dimensional stance is insufficiently contrasted with well-established categorical or staging models (e.g., MCI or prodromal stages). While this study does not center on AD, engaging with these frameworks would better position its contribution within the broader aging literature.
3. Memory tasks across datasets vary substantially in content, modality, and sensitivity. While PCA offers a practical solution to derive a general memory component, it may blur construct specificity and inflate associations with widespread brain regions. Excluding tasks with distinct modalities (e.g., visual vs. auditory memory) or conducting modality-specific analyses would help sharpen the interpretation of results.
4. Although the study targets aging, individuals in their 20s and 30s are included. Their inclusion may influence findings, particularly moderation effects (e.g., in APOE ϵ 4 carriers). Given that most participants fall within the middle-to-old age range, excluding younger individuals may yield more age-relevant insights. Relatedly, a significant portion of the sample lacks APOE genotype data. Comparing demographic and cognitive profiles of those with vs. without APOE data could help evaluate potential selection bias.
5. Despite an emphasis on nonlinearity, the estimation of change (Δ brain, Δ memory) is based on linear slopes. This introduces tension between conceptual framing and operational modeling. The authors should discuss how this choice may impact their ability to detect or interpret nonlinear patterns. Similarly, considerable analytical flexibility is evident (e.g., modeling approaches, region selection, clustering criteria), but robustness checks (e.g., multiverse or sensitivity analyses) are lacking.
6. Z-scoring based on normative modeling improves harmonization but also complicates interpretation by emphasizing relative over absolute change. Additionally, combining datasets with different inclusion criteria, demographics, and recruitment strategies introduces heterogeneity. More explicit discussion of how this affects generalizability is warranted.
7. The claim of a global brain change pattern derived from PCA (accounting for ~20.7% of variance across 19 regions) may be overstated, especially since only a subset of regions was analyzed. Conversely, the clustering results showed several regionally specific associations. The conceptual distinction between "global" and "regional" effects should be clarified.
8. The simulation models offer a creative way to interpret observed non-linear associations but are theoretical. Empirical validation, such as re-analysis using a subset of data stratified by decline severity, would bolster the argument.
9. The selection of regions presented in the main figures (e.g., hippocampus and lateral ventricles) does not consistently align with the strongest statistical associations. More rationale should be provided for featured regions. Additionally, figures and tables should adopt consistent naming and formatting (e.g., readable region names, consistent dataset abbreviations).
10. Education level, a critical proxy for cognitive reserve, is not addressed. If available, education should be included as a covariate. If unavailable, its omission should be acknowledged as a limitation.
11. Individuals with only one observation were included in preprocessing. Given potential attrition bias (e.g., differential dropout by age or cognition), justification or sensitivity analysis is needed to support this decision.

Minor Comments and suggestions

1. Clarify the criteria for identifying "brain decliners" and "maintainers."
2. Ensure consistent terminology for datasets (e.g., BETULA/UMU, BASE-II/MPIB, COGNORM/UIO) and brain regions (avoid raw FreeSurfer labels).
3. Line 210: Left hippocampus beta weight should be corrected to .165 (per Supplementary Table 1).
4. Report corresponding p-values alongside beta weights (e.g., Lines 246–248).
5. Ensure consistency in directionality and terminology of associations: main text uses "brain change–memory change," whereas figure titles reverse the order.
6. Line 678: Replace "removing \approx 1 of the observations" with "removing \approx 1% of the observations."
7. Supplementary Table 5: Correct "Obs. Memory" to "Obs. MRI."
8. Avoid redundant content across manuscript and Supplementary Materials (e.g., duplicate method descriptions).
9. In Supplementary Table 1, consider sorting regions by beta value (not alphabetically).
10. Improve readability of region names in all figures and supplementary tables (e.g., "right inferior temporal gyrus" instead of "rh_G_temporal_inf").
11. Improve subheadings in Results/Discussion to avoid repetitive phrasing.
12. Address minor typos (e.g., "Dsccovery," "were," "your predicted variable").

Reviewer comments:

Reviewer #1

(Remarks to the Author)

The authors have done a very good job answering the questions from the reviewers in a thorough and thoughtful way. This reviewer does not have anything else to add.

Reviewer #2

(Remarks to the Author)

The authors have addressed the concerns I raised, and the revisions satisfactorily resolve the issues.

Editorial comments

Note I: This document contains the author's responses to the reviewer's comments on manuscript ID NCOMMS-25-19748 entitled *Vulnerability to memory decline in aging – a mega-analysis of structural brain change*.

Note II: Editor and reviewers' comments are written in black bold caption, while the authors' responses appear in green beneath each comment. Amendments in the manuscript are also included in green italic font.

Note III: Alongside this document, we provide the revised manuscript and supplementary information as well as clean copies of both, without the tracked changes and line numbers.

Note IV: Reference numbering corresponds to this document, including citations within amended text excerpts.

1) Nature journals have recently announced an update to our guidance on reporting on sex and gender in research studies (see here). We strongly encourage researchers to follow the 'Sex and Gender Equity in Research – SAGER – guidelines'

The revised manuscript follows the recommendations from the sex and gender reporting guidelines. Self-reported nature of biological sex is now included in the revised manuscript (**p26, line 635**).

2) All Nature Communications manuscripts must include a "Data Availability" section (...) [and] a "Code Availability" section (...). 3) To maximise the reproducibility of research data, we ask that you provide a Source Data file containing the raw data underlying the following types of display items

The manuscript includes a data availability section and a code availability section following the Nature portfolio policies. We additionally include a Supporting app where complete stats are available, as well as visualization for all analyses. We cannot provide individual-level data. Each dataset has different owners (see **Supplementary Table 7**). The signed data agreements do not allow us to share individual-level data.

Reviewer #1 comments

The manuscript is well written and the authors have done an impressive work combining data from multiple studies. The authors further did a careful job explaining what they did, motivating their decisions for the statistical analyses, and discussing strengths and limitations of their methods. The findings seem robust and the conclusions are supported by the results. (...). It would benefit the manuscript even further to provide some more detail on how to interpret the findings.

We thank the reviewer for the positive evaluation of the manuscript. The reviewer has raised several important comments. We have addressed them in the revised manuscript and believe the manuscript is substantially improved. Note that we have numbered the reviewers' concerns to facilitate comprehension of this document.

1) How much of the variance in episodic memory change is explained by change in brain structure. Could the authors provide some further information on the effect size for the overall effect?

Thank you for this question. We recognize the importance of reporting variance explained and other measures of effect size, and we considered this carefully in our analyses. Ultimately, we chose to present standardized β estimates in the main text, as these are directly interpretable, and, in relatively linear models, their square approximates the proportion of variance explained. We did not emphasize variance explained in the main text for two main reasons: Non-linearity and age dependence. The brain-memory associations are non-linear and strongly age-dependent. Point estimates, reported in the main text and provided in full in the Supporting app, are therefore more informative than a single model-wide variance explained value. Such values are only directly comparable to studies with similar characteristics (e.g., age distribution, measurement reliability, non-linear modeling approach, and inclusion of cognitively unimpaired observations). In contrast, standardized point estimates can be used more broadly as reference values at a desired mean age and degree of brain change. Methodological limitations in GAMMs. In *mgcv*, there is no perfect way to estimate the variance explained by a specific regressor when both random effects and smoothing terms are included. One possibility is to approximate a partial- R^2 for the regressors of interest. In our analyses, this corresponds to variance in memory decline explained uniquely by brain changes, whereas in the age \times brain models, it reflects variance explained by the age \times brain interaction term and for the joint variance of brain change and the interaction term. Note that this estimation likely underestimates the variance explained due to: i) *mgcv*'s default variance explained metric penalizing model complexity, ii) re-smoothing in the reduced model, particularly in the age \times brain models where $s(\text{age})$ is re-fitted, and iii) variance attributable to the regressors of interest being partially absorbed by re-fitting the random-effect components. In any case, variance explained provides important information, even if the numbers cannot be taken at face value. Across models, variance explained is modest, and we consider it important to make this explicit. Selected variance explained values are now described in the revised manuscript, in the associated Supplementary Tables, and provided in full in the Supporting app. We have also clarified the use of standardized β -coefficients throughout the text.

In p30, lines 761-68. Note that the β coefficients are standardized and, therefore, somewhat comparable to an effect size measure. We estimated the partial variance explained by brain change and by brain change \times age interaction as the difference in variance explained between the full model and a null model without the regressors of interest. This approximation likely underestimates the variance explained by the regressors as i) mgcv's default variance explained metric penalizes model complexity, ii) smooth terms are re-fitted in the null model, and iii) variance attributable to the regressors may be partially absorbed by re-fitting the random-effect components.

In p8, lines 208-13. Left and right hippocampus ($\beta_{\text{weighted}[w][l]} = .165$, $p_{\text{FDR}} < 0.001$, $\text{partial-}[p]R^2 = 1.9\%$); $\beta_w[r] = .168$, $p_{\text{FDR}} < 0.001$, $\rho R^2 = 1.6\%$), left amygdala ($\beta_w = .155$, $p_{\text{FDR}} < 0.001$, $\rho R^2 = 1.1\%$), left thalamus ($\beta_w = .135$, $p_{\text{FDR}} = 0.03$, $\rho R^2 = 0.8\%$), right long insular gyrus ($\beta_w = .135$, $p_{\text{FDR}} = 0.02$, $\rho R^2 = 0.8\%$) and left parahippocampal gyrus ($\beta_w = .131$, $p_{\text{FDR}} = 0.02$, $\rho R^2 = 0.8\%$) were amongst the regions showing strongest change – change associations in above-average brain decliners.

In p10, lines 234-40. For 7 regions, age significantly moderated the change - change associations ($p_{\text{FDR}} < .05$) (Figure 2a) namely left ($p_{\text{FDR}} = 0.02$, $\rho R^2 = 1.0\%$) and right hippocampus ($p_{\text{FDR}} = 0.02$, $\rho R^2 = 1.0\%$), right inferior lateral ventricle ($p_{\text{FDR}} < 0.001$, $\rho R^2 = 1.5\%$), left lateral ventricle ($p_{\text{FDR}} = 0.05$, $\rho R^2 = 0.8\%$), right caudate ($p_{\text{FDR}} = 0.03$, $\rho R^2 = 0.6\%$), right putamen ($p_{\text{FDR}} = 0.03$, $\rho R^2 = 0.7\%$), and left short insular gyrus ($p_{\text{FDR}} = 0.02$, $\rho R^2 = 0.7\%$). Combining the variance explained by the brain change and the brain change \times age interaction regressors for the left hippocampus explained up to 2.9% of the variance in memory change.

In p29 SI, lines 623-4, caption to Supplementary Table 1. Partial- R^2 = Difference between (adjusted) variance explained by the full model and a null model without the regressors of interest.

In p30 SI, lines 629-31, caption to Supplementary Table 2. Partial- R^2 = Difference between (adjusted) variance explained by the full model and a null model without the regressors of interest: left estimate includes the brain change and the brain change \times age regressors, while the right estimate includes only the interaction R^2 .

2) Similarly, how big proportion of the association is explained by regional vs global brain atrophy?

The revised manuscript shows the variance explained by both models and the increase in variance explained by the regional compared to the global brain atrophy model.

In p12, lines 294-6. The regional model, including brain change in the eight regional clusters, explained 1.3% more variance in memory decline than the model including only the global pattern of brain decline (partial- $R^2 = 3.1\%$ and 1.9% , respectively).

3) Could the authors elaborate a bit more on what other factors may underly episodic memory change (i.e. the variance that is not explained by brain atrophy)? Which future studies should be conducted in order to answer that question?

Thank you for this interesting question. First, it's important to acknowledge that individual estimates of memory decline are inherently noisy, which places a theoretical upper bound on the variance any predictor—such as brain atrophy—can explain. Additionally, brain atrophy itself is measured with error, further limiting its observed association with memory decline. In other words, the latent (true) relationship between brain and memory change is likely stronger than what can be empirically captured in the available datasets. That said, a substantial portion of the variance in memory decline among community-dwelling individuals remains unexplained. Several plausible contributors include cortical surface area change, white matter microstructure, and vascular-related factors, including white matter hyperintensities - though they may affect other cognitive domains disproportionately. These features are at least partially independent of cortical thinning and subcortical volume loss and have been linked to cognitive outcomes in aging. It is also important to consider biological mechanisms such as tau or TDP-43 accumulation, which may partially mediate the association between structural brain atrophy and memory decline (see also **reviewer 1, comment #4**). Addressing these questions will require multimodal studies, which are logistically and analytically demanding, ideally with complete neuropathological assessment, though having enough sample size and modality breadth remains a challenge. The revised manuscript incorporates a discussion of these alternative contributors and outlines key directions for future research.

In p24, lines: 589-611. Overall, brain atrophy accounts for only a small proportion of the variance in memory decline, unsurprising given the substantial heterogeneity influencing interindividual differences in cognitive aging^{1,2}. Other brain structure modalities, such as cortical area, are both associated with memory decline and independent from cortical thinning^{3,4}. White matter integrity⁵, particularly within limbic pathways⁶, along with white matter hyperintensities⁷ and other vascular factors, are also likely to contribute, at least partially independently. Additional potential contributors include brainstem, basal ganglia, and cortical neurochemistry^{8,9} and indices of brain function¹⁰. Associations between memory decline and early pathological changes in limbic-predominant age-related TDP-43 encephalopathy (LATE-NC), primary age-related tauopathy (PART), or AD may be largely mediated by shared variance with temporal brain atrophy¹¹⁻¹³. Even accounting for these factors, substantial variance in cognitive decline will remain unexplained due to measurement error, technological limitations, and resilience factors. (...). Multimodal integration of longitudinal neuroimaging and cognitive data with postmortem neuropathology offers a promising avenue for advancing our understanding¹⁴.

4) Given that the study populations were restricted to cognitively unimpaired participants. Would the authors agree that their conclusions would mainly apply to normal aging and it is possible that the patterns would be different in a sample including also individuals who are close to a dementia diagnosis? Or do the authors think that the “brain decliners” are in fact individuals in close proximity to dementia?

Thank you for this important question. In human aging research, there is ongoing tension between theoretical frameworks rooted in neuropathology and research approaches focused on community-based samples. Our study focuses on the latter and is guided by symptomatology rather than

biomarkers or postmortem pathology. As a result, our approach imposes some limitations on generalizability and comparability to studies with biomarker-based inclusion or even other symptomatology criteria. We would like to clarify an important methodological point: we excluded specific observations, not entire participants, based on the presence of mild cognitive impairment (MCI) or dementia. This detail has some implications. 1) Our sample includes individuals for whom we have information who are close to cognitive impairment, as at later timepoints were excluded if MCI/AD was present. 2) This approach avoids additional healthy bias by retaining earlier data points from those on a declining trajectory. 3) It ensures consistency across datasets, as most participants lack prospective clinical diagnosis due to dropout or study completion, though some of them must have developed MCI shortly after their last visit. As such, while we know some participants progressed toward cognitive impairment, such information is not uniformly available across the sample.

Given these choices, we believe our findings are largely applicable to what is often referred to as *normal aging*, though we avoid using this term due to its conceptual ambiguity. Including observations from individuals with MCI would likely alter the results in predictable ways. Specifically, we would expect: 1) stronger associations between brain and memory change; 2) a more pronounced age \times brain atrophy interaction predicting memory decline; and 3) stronger effects of APOE ϵ 4 on brain and memory trajectories. These expectations align with established links between age, APOE ϵ 4, brain atrophy, and cognitive impairment. This is discussed in the revised manuscript.

A further important question is how of the observed brain-memory association can be attributed to specific factors. To meaningfully mediate this association, a pathological factor needs to 1) explain variance in brain atrophy, 2) explain variance in memory decline, and 3) be sufficiently prevalent in community-dwelling older adults. Amyloid- β (A β) is common in aging populations but is often weakly related to both brain atrophy and memory decline. Tau accumulation may be a better candidate, given its strong associations with both brain atrophy and cognitive decline when measured with PET. However, its prevalence in cognitively unimpaired individuals remains relatively low – e.g., 2.8% at age 60 and 13.2% at age 80 in a largely memory clinic-based sample - limiting its explanatory power in our study. Other plausible contributors include TDP-43 pathology and vascular changes, but at most, they will explain a limited proportion of our study findings. The revised manuscript discusses these considerations in greater detail.

In p19, lines 440-2. However, the change – change associations and the moderating effects of age were not influenced by APOE ϵ 4 status, nor are they likely affected by preclinical AD. First, in some regions, the change – change associations were evident before age 60, when prevalence of Tau deposition is generally very low¹⁵.

In p24, lines: 600-9. The relationships among these factors, as well as our findings, are shaped by sample characteristics; accordingly, our results are most generalizable to other samples of clinically unimpaired individuals. Inclusion of individuals with concurrent cognitive impairment may produce modest quantitative shifts - such as stronger effects of brain atrophy, age \times brain atrophy, and APOE ϵ 4 on memory decline – consistent with established links among these measures and cognitive impairment. In contrast, qualitative changes, such as a reconfiguration of the regional pattern or its interaction with APOE ϵ 4, are unlikely unless the sample is substantially clinically enriched. Only with significant pathology would a fundamental shift in the neurobiological basis of memory decline be expected.

5) Especially regarding the conclusion regarding the $\epsilon 4$ carriers, would the authors expect the same pattern in a sample where many are expected to develop dementia in the near future?

As briefly outlined in the previous response (**reviewer #1, comment #4**), we believe that including follow-up observations from individuals with MCI could impact some findings related to APOE $\epsilon 4$. Specifically, we would expect stronger effects of APOE $\epsilon 4$ status on brain atrophy, memory decline, and its interaction with age. This expectation is grounded in the fact that APOE $\epsilon 4$ is the strongest genetic risk factor for late-onset Alzheimer's Disease (AD). Carriers are likely to be overrepresented amongst individuals converting to MCI and tend to exhibit steeper rates of brain and memory decline.

Relaxing the inclusion criteria to incorporate prospective MCI cases is unlikely to influence the APOE $\epsilon 4 \times$ brain atrophy interaction as a predictor of memory decline. This would likely require a shift in the neurobiological basis of memory decline – one in which APOE $\epsilon 4$ carriers show a more medial temporal-centric pattern, while non-carriers show a more heterogeneous set of contributing factors. We are skeptical that this neurobiological divergence would emerge simply by including follow-up MCI observations from the current cohort, as the current patterns would still dominate. We believe this shift may occur in clinically enriched samples (e.g., including only individuals with cognitive impairment). It is more likely that in these samples, the individuals show more distinct atrophy profiles, reflecting divergent pathological profiles and thus shifting the neurobiological basis of memory decline. This issue is now discussed in more detail in the revised manuscript together with the previous **reviewer #1** comment (**#4**).

6) Introduction, line 178: what do the authors mean by the term “at-risk individuals”? They seem to be referring to individuals who decline faster than average but the term becomes circular. Thank you for pointing this out. The term was vague and poorly defined. The sentence reads more clearly without it, and it has been removed in the revised manuscript.

In p7, lines 175-6. Are change-change associations more pronounced in individuals with above-average brain decline and/or carriers of the APOE $\epsilon 4$ allele?

7) Methods: the study investigated 166 cortical and subcortical brain regions. Why were so many regions included? How were they selected?

The 166 regions include 18 subcortical areas – based on the *aseg* subcortical atlas - and 74 cortical regions per hemisphere from the *Destrieux* atlas. The *Destrieux* atlas is a widely used, morphology-based parcellation that assigns labels based on identifiable cortical sulci and gyri. We consider this atlas to be at least as well-suited to our study goals as other widely used alternatives (e.g., Desikan, Yeo, AAL, Gordon, Schaefer, etc.). Ultimately, our choice was guided by practical considerations as i) both atlases are readily available in FreeSurfer and, more importantly, ii) high-quality normative models already exist for these parcellations. Developing new normative models is time- and resource-

intensive and is typically not recommended unless substantial improvements are expected. The revised manuscript now provides additional details on the parcellations used, the rationale for atlas selection, and a discussion of atlas choice within the context of analytical flexibility.

In p22, lines 554-6. Atlas choice was guided by practical considerations, particularly the availability of high-quality normative models¹⁶. While it is unclear which atlas would be optimal for our study, a multi-modal atlas may be a promising candidate¹⁷.

In p27, lines 679-82. Data was tabulated based on the Destrieux (cortical; N = 74 regions per hemisphere)¹⁸ and aseg (subcortical, N = 18 regions) atlases¹⁹. Atlas choice was guided by practical considerations, as high-quality normative models were readily available¹⁶.

Reviewer #2 comments

(...). This study demonstrates considerable effort in integrating multisource datasets with extensive data cleaning, processing, and modeling. However, several major and minor issues warrant clarification, refinement, or additional analysis.

We thank the reviewer for the positive evaluation of the manuscript. We have addressed the concerns in the revised manuscript and believe it clearly improved.

Major Comments

1) The manuscript proposes a dimensional view of aging and brain decline, yet categorically separates individuals into "above-average decliners" and "maintainers." This introduces theoretical inconsistency. The authors should reconcile these approaches and consider aligning their framework with cognitive reserve, compensation, or resilience theories.

We understand the reviewer's point. The terms *brain decliner* and *brain maintainer* are used purely for communicative purposes. Our analyses were fully dimensional, and the individual point estimates are available in the **Supporting App**. The *brain decliner* and *brain maintainer* labels do not reflect a hard statistical threshold but instead offer an accessible shorthand for referring to individuals with relatively higher or lower levels of brain decline compared to their age and sex peers. The revised manuscript clarifies the communicative intent of these terms. See **review #2, concern #12** for a similar concern.

The manuscript exemplifies the core principles of brain maintenance theory, which posits that individuals who better preserve brain structure, brain function, or neurochemistry will exhibit less age-related cognitive decline. The optimal scenario for testing this theory involves using longitudinal data to assess whether reduced change in specific brain features is associated with more stable cognitive performances over time. This ideal design aligns with our study design and primary findings. The revised version of the manuscript acknowledges the consistency of our approach and results with the brain maintenance theory.

Cognitive reserve, compensation, and resilience theories typically involve a triad of elements: a cognitive measure, a *neurodegenerative or pathological* factor known to *directly* influence cognition, and a putative mechanism - often brain function or a sociodemographic proxy - accounting for the imperfect association between brain and cognition. The current paper does not incorporate the last element and therefore does not directly test these theoretical frameworks. The exception is the inclusion of education as a potential moderator of brain-memory, based on **Reviewer #2, comment #10**. However, we did not find supporting evidence of education as a proxy for cognitive reserve in our analyses (see response to **reviewer #2, concern #10** for details). Finally, while brain-memory decline associations are statistically significant, they explain only a modest proportion of the variance. Measurement error in estimates of brain and memory change imposes a theoretical upper limit to the proportion of variance explained. Nevertheless, a substantial portion of the variance in memory decline remains unexplained. The revised manuscript discusses additional factors – including compensatory or resilience-related mechanisms – that may account for this unexplained variance.

*In p7, lines 184-90. Throughout the text, we refer to individuals with above- or below-average brain decline – relative to their age and sex peers – as brain decliners and maintainers. These labels do not reflect strict methodological criteria but are used solely for communicative clarity, as our analyses employ a dimensional approach throughout. Finally, we provide complete statistics, point estimates, and visualizations in a **Supporting App** (https://vidalpineiro.shinyapps.io/brain_mem_change/) and simulate data to aid the interpretation of results.*

In p8, lines 204-7. The relationship for all these regions was non-linear, generally showing an association between Δ brain and Δ memory only when Δ brain was steeper than average (henceforth in brain decliners). When brain decline was milder than average (henceforth in brain maintainers), (...)

*In p23, lines 584-7. For communication purposes, we used the terms brain maintainer and brain decliner despite employing a dimensional approach. This choice favors clarity over strict precision. For a more precise representation, readers are referred to the **Supporting app**, where point estimates are provided across varying levels of brain atrophy.*

In p18, lines 426-8. These results likely underpin other categorical distinctions in neurocognitive aging²⁰ and are closely aligned with brain maintenance theory predictions²¹.

In p24, lines 599-600: Even accounting for these factors, substantial variance in cognitive decline will remain unexplained due to measurement error, technological limitations, and resilience factors.

2) In a similar vein, the dimensional stance is insufficiently contrasted with well-established categorical or staging models (e.g., MCI or prodromal stages). While this study does not center on AD, engaging with these frameworks would better position its contribution within the broader aging literature.

Thank you for this important question. In human aging research, some theoretical frameworks are rooted in neuropathology while others focus on community-based samples. Our study focuses on the latter and is guided by symptomatology rather than biomarkers or postmortem pathology. As a result, our approach imposes some limitations on generalizability and comparability to studies with biomarker-based inclusion criteria, as would be the case the other way around. We believe our findings are largely applicable to what is often referred to as *normal aging* and thus may not be generalized to MCI or dementia. Including MCI patients may alter the results in predictable ways, such as 1) stronger associations between brain and memory change; 2) a more pronounced age \times brain atrophy interaction predicting memory decline; and 3) stronger effects of APOE ϵ 4 on brain and memory trajectories. These expectations align with established links between age, APOE ϵ 4, brain atrophy, and cognitive impairment. This is discussed in the revised manuscript.

A further important question is how of the observed brain-memory association can be attributed to specific AD pathology. To meaningfully mediate this association, a pathological factor needs to 1) explain variance in brain atrophy, 2) explain variance in memory decline, and 3) be sufficiently prevalent in community-dwelling older adults. Amyloid- β ($A\beta$) is common in aging populations but is often weakly related to both brain atrophy and memory decline. Tau accumulation may be a better candidate, given its strong associations with both brain atrophy and cognitive decline when measured

with PET. However, its prevalence in cognitively unimpaired individuals remains relatively low – e.g., 2.8% at age 60 and 13.2% at age 80 in largely memory clinic-based samples – limiting its explanatory power in our study. The revised manuscript discusses these considerations in greater detail.

In p19, lines 438-42. However, the *change – change* associations and the moderating effects of age were not influenced by APOE ϵ 4 status, nor are they likely affected by *preclinical AD*. First, in some regions, the *change – change* associations were evident before age 60, when prevalence of Tau deposition is generally very low¹⁵.

In p24, lines: 589-611. Overall, brain atrophy accounts for only a small proportion of the variance in memory decline, unsurprising given the substantial heterogeneity influencing interindividual differences in cognitive aging^{1,2}. Other brain structure modalities, such as cortical area, are both associated with memory decline and independent from cortical thinning^{3,4}. White matter integrity⁵, particularly within limbic pathways⁶, along with white matter hyperintensities⁷ and other vascular factors, are also likely to contribute, at least partially independently. Additional potential contributors include brainstem nuclei neurochemistry⁸ and indices of brain function¹⁰. Associations between memory decline and early pathological changes in limbic-predominant age-related TDP-43 encephalopathy (LATE-NC), primary age-related tauopathy (PART), or AD may be largely mediated by shared variance with temporal brain atrophy¹¹⁻¹³. Even accounting for these factors, substantial variance in cognitive decline will remain unexplained due to measurement error, technological limitations, and resilience factors. The relationships among these factors, as well as our findings, are shaped by sample characteristics; accordingly, our results are most generalizable to other samples of cognitively unimpaired individuals. Inclusion of individuals with concurrent cognitive impairment may produce modest quantitative shifts - such as stronger effects of brain atrophy, age \times brain atrophy, and APOE ϵ 4 on memory decline – consistent with established links among these measures and cognitive impairment. In contrast, qualitative changes, such as a reconfiguration of the regional pattern or its interaction with APOE ϵ 4, are unlikely unless the sample is substantially clinically enriched. Only with significant pathology, would a fundamental shift in the neurobiological basis of memory decline be expected. Multimodal integration of longitudinal neuroimaging and cognitive data with postmortem neuropathology offers a promising avenue for advancing our understanding¹⁴.

3) Memory tasks across datasets vary substantially in content, modality, and sensitivity. While PCA offers a practical solution to derive a general memory component, it may blur construct specificity and inflate associations with widespread brain regions. Excluding tasks with distinct modalities (e.g., visual vs. auditory memory) or conducting modality-specific analyses would help sharpen the interpretation of results.

We agree that this is a good point, and we therefore re-estimated memory slopes using only verbal memory tests, as this was the only modality with sufficient data across datasets to support a modality-specific analysis. The resulting slopes were highly correlated with the original memory change estimates ($r = .99$), and therefore, we considered the analysis redundant. Still, as we agree with the reviewer that this is an interesting point, the revised manuscript acknowledges the lack of modality-specific analysis for memory and notes that the current estimates largely reflect verbal memory.

In p23, lines 564-8: Memory function was harmonized independently within each dataset, and thus reflects the specific tests used rather than a common construct; most of these tests primarily assessed verbal episodic learning and free recall. While modality or content-specific analyses could, in principle, reveal more precise specific anatomical relationships, the available data did not permit such detailed investigation.

4) Although the study targets aging, individuals in their 20s and 30s are included. Their inclusion may influence findings, particularly moderation effects (e.g., in APOE ϵ 4 carriers). Given that most participants fall within the middle-to-old age range, excluding younger individuals may yield more age-relevant insights. Relatedly, a significant portion of the sample lacks APOE genotype data. Comparing demographic and cognitive profiles of those with vs. without APOE data could help evaluate potential selection bias.

Thank you for the comment. We have reanalyzed the sample using individuals aged 60 years or older at baseline. In short: a) APOE ϵ 4 status effect on memory change and brain change is stronger, as expected, since the deleterious effects of APOE ϵ 4 status on cognition seem mostly constrained to older ages. b) Age \times APOE ϵ 4 status interaction on brain and memory decline disappeared due to a smaller sample age range. As in the original analysis, we do not find interaction effects of APOE ϵ 4 \times brain atrophy nor APOE ϵ 4 \times brain atrophy \times age on memory decline. All stats are included in the **Supporting App**. The revised manuscript summarizes the findings of this re-analysis. The only relevant variable associated with missing APOE data is dataset. Some datasets show higher likelihood of having missing data. Age, sex, and cognition were not related to APOE missingness. Since APOE data is missing completely at random (controlling for dataset), we decided not to take further action in the revised manuscript.

*In p15, lines 345-58. We conducted two sensitivity analyses: (...). 2) We reran the APOE ϵ 4 analyses in a subsample of individuals aged 60 years or more at baseline ($n = 2048$). As expected, the effect of APOE ϵ 4 status on memory decline was found in a greater number of brain regions. However, due to the restricted age range, APOE ϵ 4 \times age interactions were no longer significant. Consistent with the main analyses, APOE ϵ 4 status did not moderate the relationship between brain change and memory change nor its interaction with age ($p_{FDR} > .15$) (SI; **Supplementary Figure 5**). See complete statistics and additional visualization for all APOE ϵ 4 analyses in the **Supporting App**. Altogether, the results show that cognitively healthy carriers of the APOE ϵ 4 allele have steeper rates of brain and memory decline, specifically in old adulthood, but no evidence of stronger Δ brain – Δ memory associations. The regional associations between brain change and memory change exist independently of an increased presence of pathological processes, and of cognitive changes associated with genetic risk of AD.*

In p15 SI, lines 363-76. We conducted a sensitivity analysis to verify whether APOE ϵ 4 carriers show steeper rates of brain and memory decline in old adulthood, without evidence of stronger Δ brain – Δ memory. This analysis was restricted to individuals aged 60 years or more at baseline with APOE ϵ 4 status ($n = 2048$). APOE ϵ 4 status was associated with steeper memory decline ($\beta = -0.117$, $t(2069) = -$

2.9, $p = 0.004$; $F = 1.490$, $edf = 1$, $p = .50$) and with regional brain decline in 37 regions ($p_{FDR} < 0.05$). The strongest associations were observed in the left and right hippocampus ($\beta[l] = -0.209$, $\beta[r] = -0.216$), followed by ventricular regions, the left transverse occipital gyrus, and the right amygdala (**Supplementary Figure 5**). No APOE $\epsilon 4$ status \times age interactions on brain decline were significant ($p_{FDR} = 1$), including the left and right hippocampus at the uncorrected level ($p[l] = .34$, $p[r] = .37$). APOE $\epsilon 4$ did not significantly moderate the association between brain decline and memory decline, nor its interaction with age ($p_{FDR} > .2$). This lack of moderation persisted for the hippocampi at the uncorrected level ($p_{unc} > .2$). See complete statistics in the **Supporting App**. These sensitivity findings are consistent with the main APOE $\epsilon 4$ status results.

In p27 SI, lines 597-601, caption to Supplementary Figure 5. APOE $\epsilon 4$ associations with brain change – older subsample. Association between APOE $\epsilon 4$ (carriers, non-carriers) and brain change in a subsample of individuals aged ≥ 60 years with available APOE $\epsilon 4$ status. Estimates (β) for the effect of APOE $\epsilon 4$ status on Δ brain with orange color line representing $p_{FDR} < 0.05$. Δ brain represents atrophy in subcortical and thinning in cortical regions. See **Supporting app** for complete stats.

Supplementary Figure 5 snippet.

5) Despite an emphasis on nonlinearity, the estimation of change (Δ brain, Δ memory) is based on linear slopes. This introduces tension between conceptual framing and operational modeling. The authors should discuss how this choice may impact their ability to detect or interpret nonlinear patterns. Similarly, considerable analytical flexibility is evident (e.g., modeling approaches, region selection, clustering criteria), but robustness checks (e.g., multiverse or sensitivity analyses) are lacking.

We agree that there is some apparent tension between the conceptual emphasis on non-linear modeling - including the non-linear operationalization at the population level – and the operational modelling of individual change, which relied on linear slopes. However, we believe this tension is much less consequential than it may initially appear. In practice, quantifying yearly individual change using quadratic terms yields results that are highly similar to those obtained with linear slopes, with the unintended consequence that models become overparameterized in individuals with two time points. An alternative approach could involve latent longitudinal SEM, which is well-suited for modeling co-

occurring non-linear change. However, this method differs conceptually from our current framework and is ill-suited to our data, as it would struggle with i) integrating multiple datasets with heterogeneous designs and subject demographics (e.g., age) and ii) addressing unpaired cognitive and brain data. Importantly, in the preprocessing stage, we removed sex-specific non-linear age trajectories, so both MRI-based atrophy and memory decline account for group-level nonlinearities. Non-linear modelling would only be necessary if one expects notable non-linear deviations with respect to the group norm, which we do not. Lastly, one needs to consider the reliability of estimating longitudinal MRI and cognitive data. Estimating non-linear trajectories with confidence is more challenging, and probably next to impossible with, e.g., three MRI observations. The reliability of longitudinal MRI data is limited as the ratio between interindividual variability in true/latent change compared to that of measurement error is mediocre. Thus, in our view, estimating non-linear trajectories at the individual level is unnecessarily reckless, with few advantages and many risks. The revised manuscript mentions this tension more thoroughly and the rationale for estimating individual change.

Regarding analytical flexibility, we explicitly acknowledge i) that multiple analytical decisions were taken and ii) that a fully exhaustive multiverse analysis is not feasible in this context. That said, we did conduct a series of targeted sensitivity/control analyses, including models adjusting for covariates (e.g., ICV, sex, education) and analyses restricted to participants aged 60 and older. These analyses consistently yielded similar patterns of association, supporting the robustness of our main findings. We have expanded our discussion of both analytical flexibility and generalizability in the revised manuscript to reflect this point.

In p23, lines 572-5. Reliable estimation of non-linear change typically requires substantially more data than linear models. Moreover, our preprocessing pipeline explicitly accounts for and removes non-linear age trends, thereby addressing a major source of non-linearity in brain and cognitive measurements.

In pp22-3, lines 528-59. A multiverse approach²² was impractical due to constraints in data availability and computational resources. Applying iterative normative modeling and generating bootstrapped p-values across 166 regions would have rendered such an approach prohibitively time-consuming. Key considerations include: (...). In any case, we conducted a series of control analyses that included additional covariates, analytical choices, and age-restricted subsamples. These analyses consistently produced similar patterns of association, supporting the robustness of our main findings.

6) Z-scoring based on normative modeling improves harmonization but also complicates interpretation by emphasizing relative over absolute change. Additionally, combining datasets with different inclusion criteria, demographics, and recruitment strategies introduces heterogeneity. More explicit discussion of how this affects generalizability is warranted.

Indeed, this is one of the expected – and to some extent, desired - consequences of normative modelling. The shift from absolute to relative change – with respect to age and sex – facilitates the interpretation of age-specific associations as captured by the age × brain atrophy interaction. This makes the results more comparable across studies with different age distributions. One can simply

examine the standardized effect sizes at specific ages. Normative modeling often outperforms other harmonization methods.

Normative modelling harmonization output scores that are normalized by an (non-linear) age- and sex- specific means and variability compared to most other harmonization methods that do account only for mean levels. Thus, assuming equal performance in removing scanner and dataset variability, normative modelling will only differ from most harmonization methods if variability is highly dependent on age or sex. Both age and sex show non-to-modest changes in variability in cortical thickness and subcortical volumes. A notable exception are ventricular features that undergo substantial increases in variability with increasing age. As a result, estimates for ventricular regions may diverge more across methods, with effects of brain atrophy on memory change associations being attenuated. The point estimates from the age \times brain atrophy interaction should be insensitive to the method and produce standardized coefficients. The revised manuscript includes a comparison between normative modeling and generalized additive mixed model-based harmonizations and discusses where and how our preprocessing method may influence the results.

In p28, lines: 694-8. *The correlation between individual change estimates derived from the normative-modelling approach and those obtained using a GAMM – as in ²³, with sex as fixed factor, age as smoothing term, and subject, scanner, and dataset as random intercepts – was very high across most regions suggesting a null-to-modest impact of this analytical choice (mean $r = .96$ across regions; see **Supplementary Figure 6**).*

In p22, lines 537-44. *Assuming similar scanner harmonization performance, normative models should yield results comparable to other methods except in cases where interindividual variability strongly depends on age or sex. This is generally not the case for cortical thickness and subcortical volumes, but ventricular features show substantial age-related changes in variability. Consequently, we expect some attenuation in brain-memory change associations for ventricles when using normative modelling. In contrast, estimates for the age \times brain atrophy interaction should be largely method-independent and reflect standardized associations.*

In p28, SI, lines 605-16, caption to Supplementary Figure 5. Individual change estimates across harmonization methods. Correlation across regions between individual change estimates derived from normative modeling and those obtained using a GAMM-based approach, as in Fjell et al.²³. The GAMM harmonization model includes sex as fixed factor, age as a smoothing term, and individual, scanner, and dataset as random intercepts. Individual change was estimated as described in the Methods, using the residuals as input data. The boxplot displays the distribution of correlations across regions: the box represents the interquartile range (IQR), and whiskers extend to $1.5 \times IQR$. Both the mean and median correlation across regions are $r = .96$. Only 5 regions show correlation $r \leq 0.90$, four of which are ventricular features. The lower agreement for ventricular regions is likely due to pronounced age-related increases in variability in ventricular volume, which is modelled in normative models but not in the GAMM-based approach.

See **Supplementary Figure 6** snippet above.

7) The claim of a global brain change pattern derived from PCA (accounting for ~20.7% of variance across 19 regions) may be overstated, especially since only a subset of regions was analyzed. Conversely, the clustering results showed several regionally specific associations. The conceptual distinction between “global” and “regional” effects should be clarified.

We agree that the term “global” can be ambiguous and have clarified its meaning in the context of our study. We disagree, however, that the PCA results are overstated. Our goal was to test whether the associations between regional brain atrophy and memory decline could be accounted for by a shared underlying component, not the overall dimensionality of brain atrophy. In this context, we believe our approach is more methodologically justified and more targeted than estimating a single, general brain factor, which might have diluted the contributions of memory-sensitive areas while emphasizing regional effects. We now make it explicit in the revised manuscript that the “global” should not be interpreted in a general sense, but rather as a summary of shared variance among regions most relevant to memory decline in our data, and as such, comparisons with “global” factors reported elsewhere may not be entirely generalizable. We refer readers to prior work on the dimensionality of aging, which generally reports higher shared variance estimates, likely due to, abovementioned methodological differences. Two key distinctions are: i) many of our selected regions are spatially non-contiguous, and spatial proximity is known to inflate inter-regional correlations due to both shared signal and measurement error; ii) we conducted a mega-analysis across multiple cohorts, unlike previous studies using a single cohort. Finally, we acknowledge that selecting regions based on statistical significance introduces a degree of arbitrariness.

In p29, lines 725-32. We tested the dimensionality by performing PCA and clustering on the regions (N = 19) showing significant Δ brain – Δ memory associations. To ensure the resulting global atrophy factor was aligned with the association of interest, we constrained it to regions linked to memory decline. As such, it may be more appropriate to interpret this factor as reflecting the common contribution of memory-sensitive areas. Since statistical significance represents a somewhat arbitrary threshold for region selection, the global factor identified here may not be fully generalizable to broader measures of global brain atrophy. This, along with the heterogeneity introduced by combining multiple cohorts, may explain the relatively modest variance explained by the global factor compared to those reported elsewhere^{3,24}.

8) The simulation models offer a creative way to interpret observed non-linear associations but are theoretical. Empirical validation, such as re-analysis using a subset of data stratified by decline severity, would bolster the argument.

Yes, we agree that the simulation models are inherently theoretical. Our intention was to provide a post-hoc explanation for the observed non-linear associations, rather than making explicit predictions on empirical tests. We believe re-analyzing the data by stratifying participants based on decline severity would add limited insight, as the GAMM analyses already characterize brain-memory change associations as a function of severity in brain atrophy.

To robustly validate the simulation, one would require near error-free empirical estimates of individual change – something unattainable with legacy data due to substantial measurement error inherent in both MRI and cognitive assessments compared to interindividual variability in change. In fact, a key motivation behind the simulations was to illustrate potential differences between observed (noisy) empirical data and the underlying *latent* associations. Many of the assumptions underlying our simulations have been explored in prior work by our group²⁵. Novel data acquisition strategies, such as cluster scanning, may help reduce measurement error to levels that would require future empirical testing of these simulations²⁶. This validation, however, would also require longitudinal data with sufficient follow-up and sample size, which will take time to accumulate. We clarified these limitations and future directions in the revised manuscript.

In p23, lines 578-81. The two-source model proposed in the discussion remains speculative, while the specific mechanisms underlying brain aging remain elusive. Our post-hoc simulation requires external empirical validation using more precise estimates of change than those available in current legacy datasets²⁵. Improved study design and advanced scanning protocols, such as cluster scanning²⁶, offer a promising avenue to enable such validation.

9) The selection of regions presented in the main figures (e.g., hippocampus and lateral ventricles) does not consistently align with the strongest statistical associations. More rationale should be

provided for featured regions. Additionally, figures and tables should adopt consistent naming and formatting (e.g., readable region names, consistent dataset abbreviations).

The revised manuscript provides a clearer rationale for the selection of regions used in visualizations. In short, we selected from among the statistically significant regions those that best illustrate representative brain-memory change trajectories, intending to offer readers a more heterogeneous view of how these associations are throughout the brain. In each case, the regions with the highest statistical significance were prioritized. We consider this approach preferable to simply selecting the top regions based on significance or effect size alone. For example, the three most significant regions in the main analysis - Left Hippocampus, Right Hippocampus, and Left Amygdala - exhibit highly similar brain change – memory change associations profiles, which may be unsurprising given they are close anatomically, are genetically correlated, have shared ontogeny, and show similar lifespan trajectories amongst other things. The revised manuscript provides criteria for region visualization. Note also that all regions can be visualized in the **Supporting App**.

*In p28, lines 702-4. Regions were selected among the statistically significant results to best illustrate the range of change-change associations. Visualization for all regions are available in the **Supporting app**.*

10) Education level, a critical proxy for cognitive reserve, is not addressed. If available, education should be included as a covariate. If unavailable, its omission should be acknowledged as a limitation.

Thanks. Educational information is available for most participants. We harmonized education level across cohorts into two categories: high and low education levels. We introduced education level in two additional analyses: brain atrophy × education level, and brain atrophy × age × education level as predictors of memory decline. Neither main nor interaction effects involving education were statistically significant after correcting for multiple comparisons. These findings are consistent with recent longitudinal studies that question the role of education in late-life memory decline, as opposed to the memory levels, nor as a key mediator of the relationship between brain atrophy and cognitive decline. A description of methods and results is available in the **Supplementary Information** of the revised manuscript, with the complete stats reported in the interactive *shinny app*.

In p8, lines 217-20. Education level did not significantly contribute either as a resilience factor, i.e., influencing the effect of education on brain and memory decline, or as a cognitive reserve factor, i.e., moderating the association between brain and memory, consistent with recent longitudinal studies on brain and memory decline^{23,27} (SI).

In p12 SI, lines 287-315. We operationalized education level by classifying individuals as having or not having, tertiary education. In UKB and BBHI, education was reported categorically, based on ISCED level 7 and ISCED level 3 nomenclature, respectively, and thus categories were merged as appropriate. The remaining datasets reported years of education. For the ADNI, BASE-II, BETULA, COGNORM, HABS, LCBC, OASIS3, and VETSA, individuals with ≥ 16 years of education were classified as having tertiary education. For preventAD and BBHI, the threshold was set at ≥ 15 years, reflecting Quebec

scholarization norms <https://www.quebec.ca/en/education/study-quebec/education-system> and the mid-20th-century Spanish education system. The AIBL dataset did not have available information on education. See an in-depth description in Fjell and colleagues²³. Also, 17 individuals from BASE-II and 19 from LCBC did not have education level. The analyses were carried out with 3559 individuals with longitudinal information. We examined the following effects of education level on brain and memory decline: 1) education level as predictor of memory decline; 2) education level as predictor of regional brain decline; 3) education level × age interaction on memory decline; 4) education level × age interaction on regional brain decline; 5) education level × regional brain decline on memory decline; 6) education level × age × regional brain decline on memory decline. In all models, dataset was included as a random intercept. Analysis 1-2 were conducted using a linear mixed-effects models as implemented in lme4, lmerTest^{28,29}. The remaining analyses were performed using GAMMs as implemented in the mgcv R-package³⁰, with statistical significance assessed via a bootstrapping approximation as described in the main text for all GAMMs.

*Neither education level nor the education level × age interaction were associated with memory decline ($\beta = 0.014$, $t(2616) = 0.39$, $p = 0.69$; $F = 0.935$, $edf = 1$, $p = .96$). After correcting for multiple comparisons, both education level and the education level × age interaction were also unrelated to brain decline ($p_{FDR} > .3$). Likewise, no significant effects were observed for the education level × brain decline nor the education level × brain decline × age on memory decline interactions ($p_{FDR} > .3$). See complete stats in **Supporting app**. The results are in line with those reported in Fjell and colleagues²³ in a largely overlapping sample using instead a main factor of global decline.*

11) Individuals with only one observation were included in preprocessing. Given potential attrition bias (e.g., differential dropout by age or cognition), justification or sensitivity analysis is needed to support this decision.

We believe including individuals with only one observation during preprocessing – alongside those with follow-up data - is a sound methodological choice as it allows for the estimation of sex-specific age trajectories in a way that is less affected by attrition bias. In contrast, restricting the preprocessing sample to individuals with follow-up data will amplify attrition bias. Our approach supports a more robust and generalizable model and helps reduce bias in within-person change. For memory function specifically, including single-observation individuals enables better modeling of test-retest effects.

That said, one must be aware that i) MRI datasets are, almost by definition, not representative of the general population, and ii) the choice to include or exclude single-observation participants during preprocessing is unlikely to meaningfully affect the longitudinal outcomes. This is because a) estimates of individual change are primarily referenced to the individuals' own data. b) The potential impact of differential attrition with age must be small as follow-up periods are relatively short (mean follow-up time 4.5 and 5.1 years for brain structure and memory function). c) all analyses control for dataset, which should soak-up most dataset-specific differences, including attrition bias. We justify our choice in the revised manuscript.

In p26, lines: 632-6: Including these individuals alongside those with follow-up data allowed for a more robust and generalizable estimation of sex-specific age trajectories - less influenced by attrition bias –

since participants with follow-up data tend to be healthier than their age peers. This approach also enabled more accurate estimation of test-retest effects in memory assessments.

Minor Comments and suggestions

12) Clarify the criteria for identifying “brain decliners” and “maintainers.”

We have clarified the criteria in the revised manuscript. The terms *brain decliner* and *brain maintainer* are used purely for communicative purposes. Our analyses employed a dimensional approach throughout, and the full range of point estimates is available in the **Supporting App**. These labels thus do not reflect strict statistical thresholds but broadly refer to individuals with above- or below-average brain decline. See **review #2, concern #1** for a similar concern.

*In p7, lines 184-90. Throughout the text, we refer to individuals with above- or below-average brain decline – relative to their age and sex peers – as brain decliners and maintainers. These labels do not reflect strict methodological criteria but are used solely for communicative clarity, as our analyses employ a dimensional approach throughout. Finally, we provide complete statistics, point estimates, and visualizations in a **Supporting App** (https://vidalpineiro.shinyapps.io/brain_mem_change/) and simulate data to aid the interpretation of results.*

*In p8, lines 204-7. The relationship for all these regions was non-linear, generally showing an association between Δ brain and Δ memory only when Δ brain was steeper than average (*henceforth* in brain decliners). When brain decline was milder than average (*henceforth* in brain maintainers), (...)*

*In p23, lines 584-7. For communication purposes, we used the terms brain maintainer and brain decliner despite employing a dimensional approach. This choice favors clarity over strict precision. For a more precise representation, readers are referred to the **Supporting app**, where point estimates are provided across varying levels of brain atrophy.*

13) Ensure consistent terminology for datasets (e.g., BETULA/UMU, BASE-II/MPIB, COGNORM/UIO) and brain regions (avoid raw FreeSurfer labels).

Thank you very much. We have ensured consistent terminology for datasets throughout the manuscript. We now avoid raw FreeSurfer labels throughout the text.

14) Line 210: Left hippocampus beta weight should be corrected to .165 (per Supplementary Table 1).

Thank you very much. We have fixed this error in the revised version of the manuscript.

15) Report corresponding p-values alongside beta weights (e.g., Lines 246–248).

The p-value of the interaction between Left-Hippocampus and age on memory decline is reported on line 233. Beta Weights in lines 246-8 do not have a specific p-value as they refer to point estimates. We clarified this in the revised manuscript.

In p10, lines, 249-51. We use the left hippocampus to illustrate these effects: the relationship between Δ brain and Δ memory in brain decliners (i.e., point estimates) is (...).

16) Ensure consistency in directionality and terminology of associations: main text uses “brain change–memory change,” whereas figure titles reverse the order.

Thank you. We revised the manuscript and ensured consistency in directionality and terminology of associations.

17) Line 678: Replace “removing \approx 1 of the observations” with “removing \approx 1% of the observations.”
Thank you. We have fixed this error in the revised manuscript.

18) Supplementary Table 5: Correct “Obs. Memory” to “Obs. MRI.”

Again, thank you. This error is now corrected.

19) Avoid redundant content across manuscript and Supplementary Materials (e.g., duplicate method descriptions).

Thanks. We have removed redundancies between the main and the supplementary texts.

20) In Supplementary Table 1, consider sorting regions by beta value (not alphabetically).
The revised manuscript orders regions by beta value in Supplementary Table 1.

21) Improve readability of region names in all figures and supplementary tables (e.g., “right inferior temporal gyrus” instead of “rh_G_temporal_inf”).

We now use the nomenclature suggested by Destrieux, which was based on standard internationally accepted nomenclature and criteria. We maintain FreeSurfer nomenclature in the tables and the interactive application to facilitate engagement and reproducibility.

22) Improve subheadings in Results/Discussion to avoid repetitive phrasing.
The revised manuscript has improved subheadings in the **Results** and **Discussion** sections.

In p7, line 192. Linking brain change and memory decline: main effects.

In p9, line 231. Age as a moderator of brain change - memory change associations.

In p13, line 308. Influence of APOE ϵ 4 status on brain change - memory change associations

In p16, line 366. Exploring mechanisms behind brain - cognitive relationships: A Post-hoc simulation study.

In p18, lines 409-10. Brain decline - memory loss associations: a generalized phenomenon or constrained to above-average brain decliners?

In p19, lines 430-1. Brain change–Memory Change Associations in APOE ϵ 4 Carriers: Distinct Decline, Shared Mechanisms?

In p20, line 461. Age is an important determinant of brain change – memory change associations.

23) Address minor typos (e.g., "Dscovey," "were," "your predicted variable").

We have addressed the manuscript for minor typos and grammar errors.

References

1. Cox, S. R. & Deary, I. J. Brain and cognitive ageing: The present, and some predictions (...about the future). *Aging Brain* **2**, 100032 (2022).
2. Boyle, P. A. *et al.* Person-specific contribution of neuropathologies to cognitive loss in old age. *Ann Neurol* **83**, 74–83 (2018).
3. Sele, S., Liem, F., Mérillat, S. & Jäncke, L. Age-related decline in the brain: a longitudinal study on inter-individual variability of cortical thickness, area, volume, and cognition. *Neuroimage* **240**, 118370 (2021).
4. Nyberg, L., Andersson, M. & Lundquist, A. Longitudinal change-change associations of cognition with cortical thickness and surface area. *Aging Brain* **3**, 100070 (2023).
5. Ritchie, S. J. *et al.* Brain volumetric changes and cognitive ageing during the eighth decade of life. *Hum Brain Mapp* **36**, 4910–4925 (2015).
6. Henson, R. N. *et al.* Multiple determinants of lifespan memory differences. *Sci Rep* **6**, 32527 (2016).
7. Boyle, P. A. *et al.* White matter hyperintensities, incident mild cognitive impairment, and cognitive decline in old age. *Ann Clin Transl Neurol* **3**, 791–800 (2016).
8. Dahl, M. J., Kulesza, A., Werkle-Bergner, M. & Mather, M. Declining locus coeruleus–dopaminergic and noradrenergic modulation of long-term memory in aging and Alzheimer’s disease. *Neuroscience & Biobehavioral Reviews* **153**, 105358 (2023).
9. Bäckman, L., Nyberg, L., Lindenberger, U., Li, S.-C. & Farde, L. The correlative triad among aging, dopamine, and cognition: current status and future prospects. *Neurosci Biobehav Rev* **30**, 791–807 (2006).
10. Mooraj, Z. *et al.* Toward a functional future for the cognitive neuroscience of human aging. *Neuron* **113**, 154–183 (2025).
11. Costoya-Sánchez, A. *et al.* Increased Medial Temporal Tau Positron Emission Tomography Uptake in the Absence of Amyloid- β Positivity. *JAMA Neurol* **80**, 1051–1061 (2023).

12. Nelson, P. T. *et al.* Limbic-predominant age-related TDP-43 encephalopathy (LATE): consensus working group report. *Brain* **142**, 1503–1527 (2019).
13. Berron, D. *et al.* Early stages of tau pathology and its associations with functional connectivity, atrophy and memory. *Brain* **144**, 2771–2783 (2021).
14. Bennett, D. A. *et al.* Religious Orders Study and Rush Memory and Aging Project. *J Alzheimers Dis* **64**, S161–S189 (2018).
15. Ossenkopppele, R. *et al.* Tau PET positivity in individuals with and without cognitive impairment varies with age, amyloid- β status, APOE genotype and sex. *Nat Neurosci* **28**, 1610–1621 (2025).
16. Rutherford, S. *et al.* Charting brain growth and aging at high spatial precision. *eLife* **11**, e72904 (2022).
17. Glasser, M. F. *et al.* A multi-modal parcellation of human cerebral cortex. *Nature* **536**, 171–178 (2016).
18. Destrieux, C., Fischl, B., Dale, A. & Halgren, E. Automatic parcellation of human cortical gyri and sulci using standard anatomical nomenclature. *Neuroimage* **53**, 1–15 (2010).
19. Fischl, B. *et al.* Whole brain segmentation: automated labeling of neuroanatomical structures in the human brain. *Neuron* **33**, 341–355 (2002).
20. Lindenberger, U. Human cognitive aging: corriger la fortune? *Science* **346**, 572–578 (2014).
21. Nyberg, L., Lövdén, M., Riklund, K., Lindenberger, U. & Bäckman, L. Memory aging and brain maintenance. *Trends Cogn. Sci. (Regul. Ed.)* **16**, 292–305 (2012).
22. Steegen, S., Tuerlinckx, F., Gelman, A. & Vanpaemel, W. Increasing Transparency Through a Multiverse Analysis. *Perspect Psychol Sci* **11**, 702–712 (2016).
23. Fjell, A. M. *et al.* Reevaluating the role of education on cognitive decline and brain aging in longitudinal cohorts across 33 Western countries. *Nat Med* (2025) doi:10.1038/s41591-025-03828-y.

24. Cox, S. R. *et al.* Three major dimensions of human brain cortical ageing in relation to cognitive decline across the eighth decade of life. *Mol Psychiatry* **26**, 2651–2662 (2021).
25. Vidal-Piñero, D. *et al.* Reliability of structural brain change in cognitively healthy adult samples. *Imaging Neuroscience* **3**, imag_a_00547 (2025).
26. Elliott, M. L. *et al.* Precision brain morphometry using cluster scanning. *Imaging Neuroscience* **2**, 1–15 (2024).
27. Lövdén, M. *et al.* No moderating influence of education on the association between changes in hippocampus volume and memory performance in aging. *Aging Brain* **4**, 100082 (2023).
28. Bates, D., Mächler, M., Bolker, B. & Walker, S. Fitting Linear Mixed-Effects Models Using lme4. *Journal of Statistical Software* **67**, 1–48 (2015).
29. Kuznetsova, A., Brockhoff, P. B. & Christensen, R. H. B. lmerTest Package: Tests in Linear Mixed Effects Models. *Journal of Statistical Software* **82**, 1–26 (2017).
30. Wood, S. N. *Generalized Additive Models: An Introduction with R.* (Chapman and Hall/CRC, 2017).